# Optimizing Language Models for Inference Time Objectives using Reinforcement Learning

Yunhao Tang [1] [*]   Kunhao Zheng [2] [*]   Gabriel Synnaeve [2]   Rémi Munos [2] [*]

## Abstract

In this work, we investigate the merits of explicitly optimizing for inference time algorithmic performance during model training. We show how optimizing for inference time performance can improve overall model efficacy. We consider generic inference time objectives with $k$ samples, with a focus on pass@$k$ and majority voting as two main applications. With language model training on reasoning datasets, we showcase the performance trade-off enabled by training with such objectives. When training on code generation tasks, we show that the approach significantly improves pass@$k$ objectives compared to the baseline method.

## 1. Introduction

Traditionally, the performance of machine learning system can be improved via either training time or inference time algorithm. At training time, knowledge is distilled into model weights via gradient descent (Bottou, 2010). At inference time, a fixed model is queried multiple times to deliver a better prediction than a single model call.

Though model training and inference time compute improves model performance, their scaling properties differ significantly. For example, Lerer et al. (2020) has demonstrated that for the game of Hex, scaling inference time compute can match performance at increased training budget, while at a fraction of the cost. More generally, inference compute has proved effective at various machine learning applications, even if the underlying model has been trained extensively. Notable examples include board game (Silver et al., 2014; 2018), competitive pokers (Brown & Sandholm, 2018; 2019), general game (Lerer et al., 2020; Bakhtin et al., 2022), competitive programming (Li et al., 2022) and language modeling (Jaech et al., 2024; Guo et al., 2025).

[*]Equal contribution  [1]Meta GenAI  [2]Meta FAIR. Correspondence to: Yunhao Tang <yt2541@columbia.edu>, Kunhao Zheng <kunhao@meta.com>.

*Proceedings of the 42nd International Conference on Machine Learning*, Vancouver, Canada. PMLR 267, 2025. Copyright 2025 by the author(s).

Usually, model training does not explicitly account for the downstream inference time algorithm. It is clear that there is a trade-off: this helps avoid premature specialization, but might also risk not benefiting from the inference time algorithm fully had we known the algorithm in advance.

In this work, we investigate the impact of training explicitly for the inference time algorithm. We focus on pass@$k$ and majority voting, two influential yet relatively simple inference time objectives. The pass@$k$ objective arises when the task has verifiers and the system can retry it $k$ times. Majority voting is even more broadly applicable: devoting more inference time compute to have different solutions and selecting the most voted one. We demonstrate how such objectives can be optimized via stochastic gradient descent, and built as part of an online reinforcement learning algorithm (Section 3). We carry out extensive ablations that showcase the trade-off of different objectives, such as an improved inference time performance when the training algorithm is aware of the inference time algorithm (Section 5): we show that when training on mathematical reasoning datasets such as MATH, as well as challenging code generation datasets such as CodeContests, new algorithmic variants achieve significant gains on inference time objectives of interest.

## 2. Reinforcement learning for language model

A language model can be understood as a policy $\pi_\theta$ in the context of reinforcement learning (RL) (Sutton & Barto, 1998). Given a prompt $x$, the policy generates a response $y$, which then gets assessed by a human user. Usually, the objective is to optimize $\pi_\theta$ such that certain reward function $r(x, y)$ that captures human preference (HF) is maximized (Christiano et al., 2017; Ziegler et al., 2019; Ouyang et al., 2022; Bai et al., 2022). Formally, consider the average reward optimization problem

$$\max_\theta \mathbb{E}_{x \sim \rho, y \sim \pi_\theta(\cdot|x)} \left[ r(x, y) \right],$$

where $\rho$ is a distribution over prompt set. Below, when the context is clear we will omit the dependency on the prompt $x$. Given samples $y$ drawn from $\pi_\theta$, we can construct stochastic gradient estimates to the policy gradient (Sutton et al., 2000) to iteratively improve the policy. Note that for

simplicity, we have omitted the regularization prevalent in the RLHF formulation (Ouyang et al., 2022).

**Inference time objectives.** At inference time when the model is deployed, one might adopt a procedure different from training time to obtain better performance. Depending on the application, different inference time objectives are desired. One important class of inference time objective is *pass@k*, where the model is given a budget of $k$ generations at test time. Though quite a lenient metric, pass@$k$ is especially useful for difficult problems (e.g., (Li et al., 2022)) that allow for trying multiple times. It also approximates best-of-$k$ assuming access to a good reward model. Formally,

$$\mathbb{E}_{(y_i)_{i=1}^k \sim \pi_\theta(\cdot|x)} \left[\max\left(r_1, \ldots, r_k\right)\right],$$

where $r_i = r(x, y_i)$ denotes the reward for generation $y_i$.

Another important example of inference time technique is *majority voting*. The model is still given a budget of $k$ generations and outputs a single solution for assessment. In general, majority voting is applicable if the generation $y = (c, a)$ can be decomposed into a chain-of-thought $c$ and a final answer $a$. The reward scores only the final answer $r(x, y) = r(x, a)$, i.e., for problems with short and verifiable answers. It has proved highly effective at leveraging inference time compute for improved performance in various domains (Wang et al., 2022; Lightman et al., 2023). Concretely, the majority voting objective is

$$\mathbb{E}_{(c_i, a_i)_{i=1}^k \sim \pi_\theta(\cdot|x)} \left[r\left(\text{maj}(a_1, \ldots, a_k)\right)\right],$$

where the operation $\text{maj}(\cdot)$ extracts the majority element from a set of items. We assume random tie breaks in case more than one element takes the majority.

We will focus on these two objectives as they are self-contained given a language model. There are alternative inference time methods such as best-of-$k$ with access to auxiliary models (e.g., reward models) (Uesato et al., 2022; Lightman et al., 2023) which we do not discuss.

# 3. Optimizing Language Models for Inference Time Objectives using RL

These inference time objectives share a key feature: they all use $k$ samples. We propose a general formulation of $k$-sample objectives that can be optimized by stochastic gradient descent.

In general, we consider the $k$-sample objective

$$\mathbb{E}_{(y_i)_{i=1}^k \sim \pi_\theta(\cdot|x)} \left[f\left(x, y_1...y_k\right)\right], \quad (1)$$

defined through a function $f : \mathcal{X} \times \mathcal{Y} \times \mathcal{Y}... \to \mathbb{R}$. Here $f$ is an aggregation function that can process an arbitrary number of generations. It is clear that both pass@$k$ and majority voting objectives are special cases.

## 3.1. Unbiased stochastic gradient estimate

To optimize for the objective in an unbiased way with stochastic gradient descent, we can sample $k$ generations i.i.d. $(y_i)_{i=1}^k \sim \pi_\theta(\cdot|x)$ and construct the gradient estimate akin to REINFORCE (Williams, 1992)

$$f\left(x, y_1...y_k\right) \sum_{i=1}^k \nabla_\theta \log \pi_\theta(y_i|x).$$

Importantly, here we *sum* over the weighted gradient of log probabilities across $k$ samples. This is in contrast to a $k$-sample policy gradient estimate that *averages* over $k$ samples. The key difference is that these $k$ samples are coupled through the aggregation function $f$. As a result, such a gradient estimate has high variance on the order of $\mathcal{O}(k)$ (Fallah et al., 2020a; Tang, 2022) that increases with the number of samples $k$, as opposed to $\mathcal{O}(k^{-1})$ in $k$-sample average policy gradient estimate.

Henceforth for notational simplicity, we let $f(\mathbf{y})$ be a short hand notation for $f(x, y_1...y_k)$ and let $f(\mathbf{y}_{-i})$ be a short hand notation for leaving out generation $i$: $f(\mathbf{y}_{-i}) := f(x, y_1...y_{i-1}, y_{i+1}...y_k)$. We also drop the dependency on $x$ when the context is clear.

For variance reduction, we propose the leave-one-out control variate that results in the following gradient estimate

$$\sum_{i=1}^k \left(f\left(\mathbf{y}\right) - f\left(\mathbf{y}_{-i}\right)\right) \nabla_\theta \log \pi_\theta(y_i|x). \quad (2)$$

The effective *advantage* function $A_i := f(\mathbf{y}) - f(\mathbf{y}_{-i})$ measures the contribution that $y_i$ has on improving from $f(\mathbf{y}_{-i})$ to $f(\mathbf{y})$.

**Lemma 1.** (**Unbiased leave-one-out gradient estimate**) The gradient estimate with the leave-one-out control variate in Eqn (2) is unbiased.

*Proof.* For any $i$, $\mathbb{E}\left[f(\mathbf{y}_{-i}) \nabla_\theta \log \pi_\theta(y_i|x)\right] = 0$ since all $k$ samples are independent, hence the proof is concluded. $\square$

Let us examine a few special cases to make concrete the concept of leave-one-out gradient estimate.

**Average reward:** $f(\mathbf{y}) = \frac{1}{k} \sum_{i=1}^k r_i$. In this case, the form of the policy gradient estimate is equivalent to the leave-one-out control variate proposed in a number of prior work that tackles $k$-sample average objectives (Kool et al.,

2019; Mnih et al., 2016).

$$\sum_{i=1}^{k}\left(\frac{1}{k}\sum_{j=1}^{k}r_j - \frac{1}{k-1}\sum_{j\neq i}r_j\right)\nabla_\theta \log \pi_\theta(y_i|x)$$

$$= \frac{1}{k}\sum_{i=1}^{k}\left(r_i - \frac{1}{k-1}\sum_{j\neq i}r_j\right)\nabla_\theta \log \pi_\theta(y_i|x).$$

**Pass@$k$:** $f(\mathbf{y}) = \max(r_1...r_k)$. With pass@$k$, the advantage function for generation $i$ reduces to $A_i = \max(r_{1:k}) - \max(r_{-i})$. In contrast to the previous example, the advantage $A_i$ here is non-negative. In fact, if we assume the reward is ordered as $r_{(1)} \leq r_{(2)} \leq ... \leq r_{(k)}$ we can rewrite the gradient estimate equivalently as

$$\left(r_{(k)} - r_{(k-1)}\right)\nabla_\theta \log \pi_\theta(y_{(k)}|x),$$

i.e., the advantage is non-zero only for the best generation $y_{(k)}$. The advantage is effectively the difference between the best and the second best reward. Such a rewrite is sensible because, if the second best and the best reward are the same, there is no incentive in updating the policy.

To impart more intuition on the sparsity of the learning signal, consider a binary reward problem for pass@$k$: we expect the learning signal to be more dense when the average solve rate is at $O(k^{-1})$. If the solve rate is higher, the problem is too easy; otherwise the problem is too hard. In both cases, the learning signal decreases for the gradient estimate.

**Majority voting:** $f(\mathbf{y}) = r(\mathbf{maj}(\mathbf{a}))$. Here, we will adopt the notation for $\mathbf{a} := a_{1:k}$ and $\mathbf{a}_{-i}$ for the leave-one-out variant. With majority voting, the advantage function for generation $i$ is $A_i = r(\mathrm{maj}(\mathbf{a})) - r(\mathrm{maj}(\mathbf{a}_{-i}))$, measuring the impact of answer $a_i$ on the majority voted answer $a$. Assume that all $m < k$ unique answers are sorted in their count $|a_{(i)}|$ as $a_{(1)} \leq ...a_{(m)}$, then $(\mathrm{maj}(\mathbf{a}_{-i}) \neq (\mathrm{maj}(\mathbf{a})$ if and only if $a_i = a_{(m)}$ and $|a_{(m)}| = |a_{(m-1)}| + 1$, since this is the only case where the leave-one-out voted answer changes. In other words, the advantage is all-zero in case a particular answer $|a_{(m)}|$ dominates the count such that $|a_{(m)}| > |a_{(m-1)}| + 1$ since there is no incentive in updating the policy.

### 3.2. Trading-off further variance reduction with bias

In general, with the leave-one-out control variate, the advantage estimate becomes more *sparse*. Indeed, the effective advantage $A_i$ measures the impact of a individual sample on the global objective. For example, for the pass@$k$, $A_i$ is only non-zero for the reward maximizing generation; for the majority voting, $A_i$ is non-zero only when it is possible to alter the majority voted answers.

Intuitively, these observations imply that the update has high variance. However in a sense, the leave-one-out control variate (Eqn (2)) marks a near optimal trade-off between simplicity and variance reduction for an unbiased gradient estimate. To reduce variance further, we can introduce bias: by subtracting the mean of all advantages, this produces a general gradient estimate

$$\sum_{i=1}^{k}\underbrace{\left(f(\mathbf{y}) - f(\mathbf{y}_{-i}) - \bar{A}\right)}_{A_i'}\nabla_\theta \log \pi_\theta(y_i|x). \quad (3)$$

where the additional baseline is $\bar{A} := \frac{1}{k}\sum_{i=1}^{k}A_i = \frac{1}{k}\sum_{i=1}^{k}f(\mathbf{y}) - f(\mathbf{y}_{-i})$. By construction, the new advantage $A_i' = A_i - \bar{A}$ is zero-mean and arguably the new estimate has even lower variance. However, such an estimate introduces a bias to the objective it optimizes.

**Lemma 2.** (**Objective of the biased gradient estimate**) The gradient estimate with zero-mean advantage function in Eqn (3) optimizes for an alternative objective

$$\mathbb{E}_{(y_i)_{i=1}^{k}\sim\pi_\theta(\cdot|x)}\left[\frac{1}{k}\sum_{i=1}^{k}f(\mathbf{y}_{-i})\right]. \quad (4)$$

*Proof.* The gradient bias comes from the baseline term which evaluates to $\mathbb{E}\left[\sum_{i=1}^{k}-\bar{A}\nabla_\theta\pi_\theta(y_i|x)\right]$. A simple algebraic manipulation shows that the aggregate gradient expectation is

$$\mathbb{E}\left[\sum_{i=1}^{k}\frac{1}{k}\sum_{j=1}^{k}f(\mathbf{y}_{-j})\nabla_\theta \log \pi_\theta(y_i|x)\right]$$

which is the unbiased gradient estimate to the objective $\mathbb{E}\left[\sum_{i=1}^{k}\frac{1}{k}\sum_{j=1}^{k}f(\mathbf{y}_{-j})\right]$. $\square$

The objective differs from the initial $k$-sample objective in Eqn (1) - it measures the leave-one-out $k-1$-sample objective $f(\mathbf{y}_{-i})$ averaged over all $k$ samples. The bias is hence

$$\mathbb{E}_{(y_i)_{i=1}^{k}\sim\pi_\theta(\cdot|x)}\left[f(\mathbf{y}) - \frac{1}{k}\sum_{i=1}^{k}f(\mathbf{y}_{-i})\right].$$

Let us characterize the bias in a few special cases. For the average reward case $f(\mathbf{y}) = \frac{1}{k}\sum_{i=1}^{k}r_i$ there is no bias. This is compatible with the fact that the advantages $A_i$ are already zero-mean. The bias is non-trivial in general.

**Pass@$k$.** Recall that the samples are ordered $r_{(1)} \leq r_{(2)} \leq ...r_{(k)}$. We can write explicitly the new objective for the biased gradient estimate in Eqn (4) as

$$\mathbb{E}\left[\frac{k-1}{k}r_{(k)} + \frac{1}{k}r_{(k-1)}\right]$$

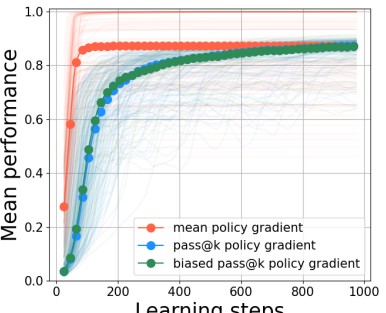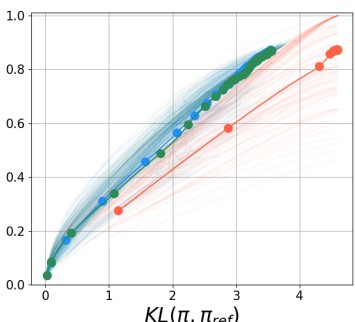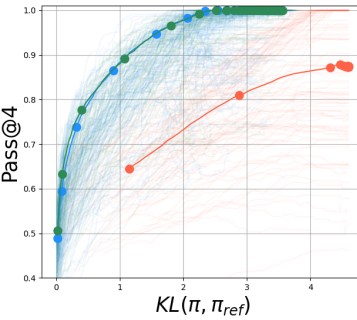

*Figure 1.* Comparison of different gradient estimates in a bandit case. We set up a bandit problem with $|\mathcal{Y}| = 100$ possible actions and each reward $r(\mathbf{y})$ is a deterministic scalar sampled from unit Gaussian. We compare three algorithmic variants: the mean policy gradient, the pass@$k$ policy gradient and its biased variant. All algorithms apply $k = 4$ samples per update with learning rate $\eta = 1.0$. Overall, we see that the baseline gradient makes the fastest improvement on the mean performance, when graphed against the learning steps (left plot); however, it is generally less KL-efficient than other $k$-sample alternatives (middle plot). When measuring the pass@$k$ performance, the $k$-sample gradient estimates lead to significantly faster improvements (right plot).

which is a convex combination of the best and the second best reward. The bias is hence $\frac{1}{k}\mathbb{E}[r_{(k)} - r_{(k-1)}]$ which is the gap between the best and the second best reward. Though such bias vanishes as $k$ increases, it also means that the overall gradient estimate has sparser signal. It is less insightful to make explicit the new objective for the majority voting case, which we detail more in Appendix B.

### 3.3. Additional discussions

We provide further discussion on the property of the $k$-sample gradient estimate.

**Interpolating mean and general $k$-sample objectives.** We see that by subtracting the mean of the $k$-sample advantage in Eqn (3), we manage to recover a smoother average of leave-one-out objective (Eqn (4)). In fact, by taking this approach this approach further and constructing leave-$n$-out objectives, we can construct an interpolation between the original $k$-sample objective and the mean objective. See Appendix B for more discussions.

**Effect of increasing number of samples $k$.** For the baseline policy gradient, varying the number of sample $k$ does not impact the expectation of the gradient estimate. The corresponding gradient estimates always approximate the gradient of the mean performance. Increasing the value of $k$ helps reduce variance of the estimate, but has a diminishing effect when the batch size is large enough. For the $k$-sample objectives, varying the value of $k$ changes the objective itself. For the pass@$k$ objective, increasing $k$ means that we care increasingly about the extreme values (i.e., max values) of the reward distribution. For the majority voting objective, as $k \to \infty$, we optimize for the margin likelihood that the final answer $a$ is matched against $a^*$. We will ablate the empirical impact of $k$ in Section 5.

**Empirical study on a bandit case.** To better illustrate the property of the $k$-sample gradient variants, we investigate the behavior of various policy gradient estimates in the bandit setting. There are a total of $|\mathcal{Y}| = 100$ actions and each incurs a deterministic reward $r(\mathbf{y})$ sampled from a unit Gaussian. We consider softmax parameterizations $\pi_\theta(\mathbf{y}) \propto \exp(\theta(\mathbf{y}))$ and all algorithmic variants apply $k = 4$ samples for the update and identical hyperparameters and initializations. Figure 1 shows the evaluated performance of different variants over time.

When measuring the mean performance, the mean policy gradient algorithm clearly maximizes the performance at a faster rate. However, when graphed against the KL divergence $\mathbb{KL}(\pi_\theta, \pi_{\text{ref}})$ against the initial policy $\pi_{\text{ref}}$, we see that the pass@$k$ gradient estimate (Eqn (2)) and its biased variant (Eqn (3)) is more *KL-efficient* (Gao et al., 2023). This follows from the fact that the $k$-sample gradient estimates have more conservative updates compared to the mean gradient. In general, maximizing the mean performance can also improve pass@$k$ performance as we observed in the plot. In this case, the other two algorithmic variants are by design more efficient.

## 4. Related work

We discuss how our approach relates to prior work.

**Other multi-sample objectives.** Multi-sample objectives have been considered extensively in the variational inference literature (Raiko et al., 2014; Burda et al., 2015; Mnih & Rezende, 2016). Using terminologies from this work, they consider $k$-sample objectives of the following form

$$\mathbb{E}\left[\log\left(\frac{1}{k}\sum_{i=1}^{k}\frac{p(x, y_i)}{q(x, y_i)}\right)\right]$$

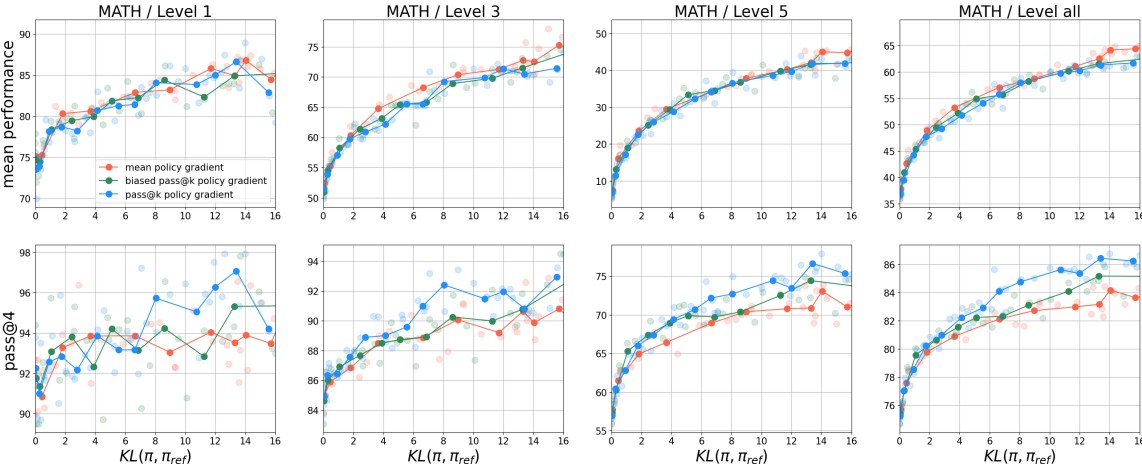

*Figure 2.* MATH training pass@$k$ 8B model. We compare three baselines: regular mean policy gradient algorithm and two variants of pass@$k$ policy gradient algorithms (unbiased and biased). We split the performance across MATH difficulty level and report the mean performance and pass@4 performance over time. We observe that as training progresses, pass@$k$ policy gradient algorithms seem to display a slight advantage over the baseline algorithm.

where $p, q$ are usually two distributions. Such an objective is by design a lower bound to the marginal likelihood, which is often considered the ultimate objective (Burda et al., 2015). In gradient-based meta-learning, the $k$-sample objective takes the following form (Finn et al., 2017; Fallah et al., 2020a;b; Tang, 2022)

$$\mathbb{E}\left[ J\left( \theta + \frac{1}{k}\sum_{i=1}^{k} g(x, y_i) \right) \right]$$

where $g(x, y_i) \in \mathbb{R}^d$ is meant to be a gradient update to the vector $\theta \in \mathbb{R}^d$ and $J : \mathbb{R}^d \to \mathbb{R}$ is a scalar function. In both cases, the effective aggregate function $f$ is a transformation of the average function. Intuitively, such objectives are *smoother* than pass@$k$ and majority voting. The specific structure of this objective also produces gradient estimate that trades-off bias with variance (Fallah et al., 2020b; Tang et al., 2021), complementary to our development here.

**Leave-one-out control variate.** The leave-one-out control variate has been studied in Mnih & Rezende (2016) for variational inference, and in Kool et al. (2019); Ahmadian et al. (2024) for improving the mean objective in a REIN-FORCE algorithm. As an alternative, Mnih & Rezende (2016) analyzed the control variate $v_i := \mathbb{E}_{\mathbf{y}_{-i}}[f(\mathbf{y})|y_i]$ which accounts for the credit assigned to other $k-1$ samples in an expected sense. Such an objective can be especially useful for the $k$-sample objectives in our case since the advantage $\tilde{A}_i = f(\mathbf{y}) - v_i$ is by construction zero-mean and arguably has lower variance. However, estimating such quantities introduces additional overhead in practice.

**Optimizing for inference time objectives.** Concurrently, Balashankar et al. (2024) proposed to optimize for pairwise win rate based on best of $k$ sampling. Under certain conditions, they showed that the objective is equivalent to the regularized RL problem with a transformed reward, which can be approximately optimized. Amini et al. (2024) pursued a similar approach but noticeably took a log transformation of the score. Such an objective is not possible to estimate in an unbiased, instead, a variational approximation is needed. Chow et al. (2024) proposed to optimize best of $k$ performance for applications where a scoring function (or verifier) is available during training. Both work have shown merits in accounting for inference time objectives during training.

## 5. Experiments on mathematical reasoning

Throughout, we focus on the mathematical reasoning dataset MATH (Hendrycks et al., 2021) where the prompt $x$ consists in asking the model a mathematical question with a short-form ground truth answer $a^*$ available. Given the model generation $y = (c, a)$ which typically consists of a step-by-step chain-of-thought reasoning $c$ and a proposed answer $a$, the reward is computed as a match between $a$ and $a^*$. We adopt Sympy (Meurer et al., 2017) to automatically match the answers and assign a reward of $r = 1$ if there is a match and $r = -1$ otherwise. As such, the objective $\mathbb{E}[r]$ also measures the average accuracy of the policy. We train on models on the MATH training set with various objective alternatives introduced above. During experiments, we provide the model with a system prompt that asks for a step-by-step solution, followed by a final answer.

Our main experiments are based on the 8B model from the

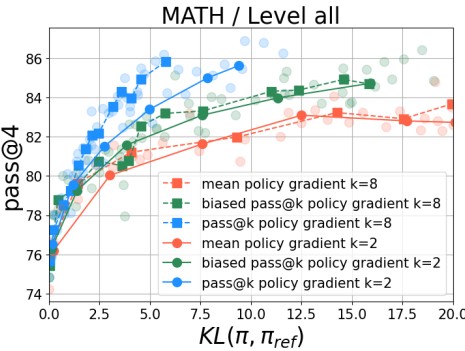

*Figure 3.* Ablation with number of samples $k$. We vary the number of samples $k$ for each gradient update for the pass@$k$ objective. We observe a more efficiency gains for the pass@$k$ gradient estimates compared to the policy gradient baseline. Importantly, note that as $k$ varies, the pass@$k$ algorithm changes its objectives.

Llama 3 model family (Dubey et al., 2024) though we also carry out ablations on models of other sizes. We apply online policy gradient algorithmic variants and investigate their performance during training. All variants apply identical hyper-parameters such as that they all apply $k = 4$ samples for gradient estimations, which we detail in Appendix A.

### 5.1. Training performance

**Pass@$k$.** Figure 2 shows the training performance of a few algorithmic variants over time. The MATH dataset contains 5 difficulty levels (the easiest is level 1 and the hardest level 5). We break down performance based on the levels and report both mean performance and pass@$k = 4$.

A few observations are in order: when measuring the mean performance, the mean policy gradient seems to obtain a slight improvement over algorithms that aim for the pass@$k$ performance. Meanwhile, the dedicated $k$-sample gradient estimates generally perform better in pass@4. The biased variant (green) to generally strike a trade-off between the pass@$k$ gradient estimate (blue) and the baseline, which is compatible with the theoretical designs. The breakdown shows that for pass@$k$, most performance difference comes from the more difficulty categories than easy ones. This might be explained by the fact that since the $k$-sample gradient estimates update policy more conservatively, they tend to have relatively more updates for the more difficult prompts, compared to the mean policy gradient.

**Majority voting.** In a similar vein, Figure 15 shows the training performance of a few algorithmic variants. We break down performance by difficulty level and measure both the mean performance and the majority voting at $k = 4$ performance. Interestingly, under this set of comparison, the policy gradient algorithm with majority voting at $k = 4$ al-

gorithm (blue) seems to obtain a better training performance overall, both in terms of the mean and the majority voting at $k$ performance. Meanwhile, the biased variant seems to perform similarly as the policy gradient baseline.

The breakdown also shows that most improvements seem to come from higher level of difficulty compared to the easier categories. This might be explained via a similar argument as the pass@$k$ algorithm: the $k$-sample gradient estimates tend to have sparser advantage and hence less update for the model over a fixed number of training steps. This makes them slightly more KL-efficient since they focus more on more challenging problems.

### 5.2. Ablations

We carry out ablations along a few dimensions of interest.

**Number of samples $k$.** We first ablate with the number of samples $k$ used for constructing the gradient estimates. Throughout, we still apply the same hyper-parameter including batch size as the base experiments. We focus on the pass@$k$ objective here and discuss more about the majority voting objective in Appendix A.

Figure 3 shows the training performance as we vary the number of samples $k \in \{2, 8\}$. Overall, we observe that as $k$ increases, there is no discernible difference in the performance of the policy gradient estimates. However, increasing $k$ seems to make the pass@$k$ sample estimates more *KL-efficient*, i.e., $k = 8$ obtains a better performance than $k = 2$ given a fixed KL budget. The performance improvement over the policy gradient estimate is also quite significant. However, though the pass@$k$ algorithm is more KL-efficient as $k$ increases, they tend to also deviate less from the reference policy $\pi_{\text{ref}}$ with a fixed number of learning steps. This means that to achieve a target level of pass@$k$ performance, the regular policy gradient baseline, despite being less KL-efficient, can still retain an advantage due to its faster deviation from the reference policy.

**Model size.** We also replicate similar experiments across a few other model sizes: 3B and 70B, both from the Llama 3 model family. We notice that 3B model tends to display a similar trend of performance trade-off compared to the 8B: the specialized $k$-sample estimates tend to outperform baselines on $k$-sample objective during training, while slightly under-performing for the mean performance. The performance improvement also mostly stems from the more difficult subset of the dataset.

The performance trade-off for the 70B case is less clear. The policy gradient baseline remains performant even for the $k$-sample objectives. We suspect that this is because the MATH dataset is a relatively simple task for the 70B model (Yue et al., 2024). Indeed, both the pass@$k = 4$ and

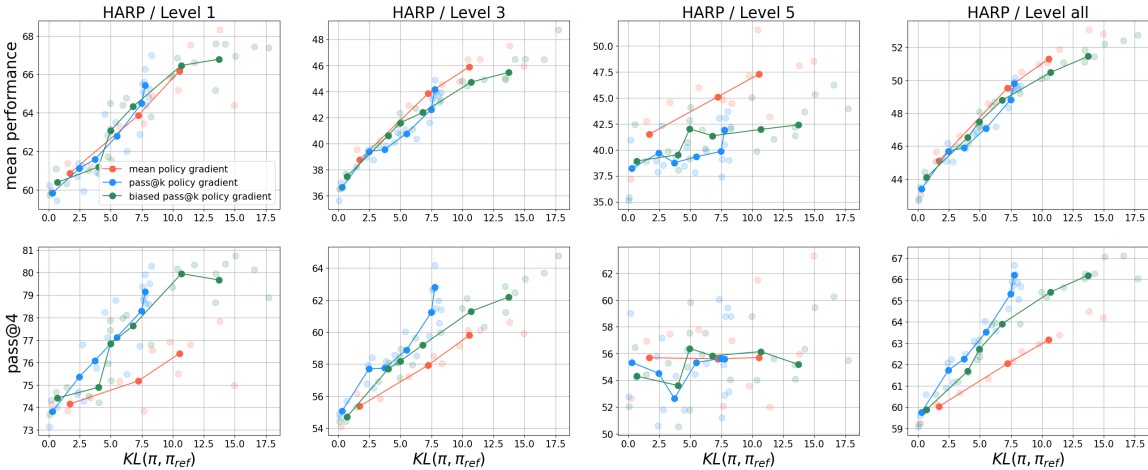

*Figure 4.* HARP training pass@$k$ 70B model. We observe that the regular policy gradient estimate improves over the pass@$k$ variants for the mean performance metric, while under-performing on the pass@$k$ objective. Such a trade-off is less significant for the MATH dataset, where we speculate that the 70B model is too powerful and learning signals are too sparse to make a difference.

majority voting at $k = 4$ are $\sim 90\%$ for the MATH training set. This implies that the $k$-sample gradient estimates will not produce much signal, given a fixed training budget.

**Training on HARP.** In light of the previous observation, we examine HARP dataset (Yue et al., 2024), a carefully curated math dataset consisting of hard problems. Though HARP was meant for evaluation, we monitor algorithmic variants' training performance as a sanity check. Figure 4 shows the results for the pass@$k$ objectives. For the pass@$k = 4$ objective, we observe a significant improvement of $k$-sample gradient estimates compared to the baseline algorithm. The mean performance of various algorithms is similar, as expected. We also observe improvement in the majority voting metric, as well as for the 8B model - though the improvement appears less significant compared to the 70B case. This might be because the HARP dataset is too difficult for 8B models. See results in Appendix A.

### 5.3. Evaluation performance

Throughout evaluation on the MATH test set, we apply a temperature sampling with $\tau = 1, \text{top p} = 1$ to be identical to the training setup. In Figure 14 we showcase the evaluation metrics for the pass@$k$ objectives. We show the majority voting metrics in Appendix A. We noted that training and evaluation performance are not perfectly correlated - we speculate this might be because since regular policy gradient estimate performs more effective updates during training, it makes up for more generalization gap from training to evaluation.

In particular, the policy gradient baseline performs quite well for the evaluation pass@4, despite being slightly outperformed by the $k$-sample pass@$k$ gradient estimates at

training time. We also note that, as training progresses, the policy gradient estimate tends to regress on pass@$k$ when $k$ is large ($k \in \{16, 64\}$), arguably due to impacts from sample diversity.

## 6. Experiments on code generation tasks

For code generation, we conduct our experiments on Code-Contests (Li et al., 2022), a competitive programming benchmark containing 13k problems as training set and evaluate on the valid and test set. Similar to previous work (Xu et al., 2024; Gehring et al., 2024), we use a $r = +1/-1$ reward indicating whether the Python code solutions pass all the given tests that come with the dataset. Due to the challenging nature of CodeContests, we use Llama 3.1 70B Instruct model and optimize for pass@8 metrics on the CodeContests training set. To measure generalization, we also report the performance on another competitive programming benchmark, TACO (Li et al., 2023) on the *easy* and *hard* split, for which we have decontaminated the CodeContests training set against TACO eval set.

Figure 5 shows a clear trade-off between optimizing for pass@1 and pass@8 on CodeContests valid and test set. This trade-off generalizes in-domain to TACO *easy* and *hard* split. Across all evaluation set, mean policy gradient achieves the best pass@1 performance (or mean performance). However, it also clearly degrades on pass@8 as the training progresses. In clear contrast, pass@$k$ policy gradient achieves the best pass@8 performance. The pass@$k$ performance improves since the onset and remains so over the course of training. We do not observe a significant difference between different pass@$k$ gradient variants, since they both achieve much better performance than the baseline.

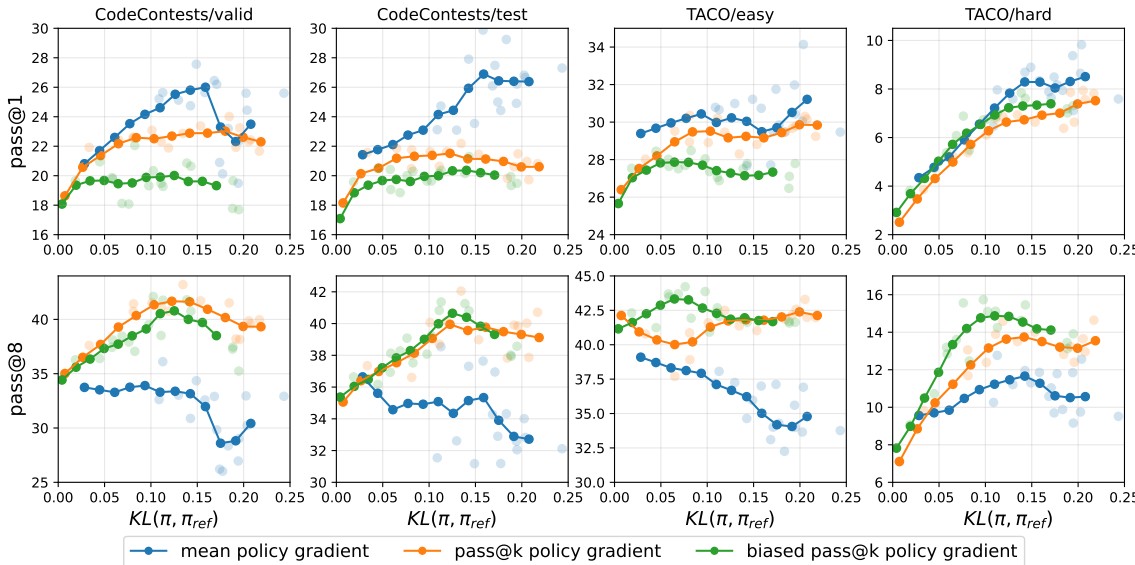

*Figure 5.* Code generation task evaluation performance. We observe a clear trade-off between pass@1 and pass@8 on CodeContests and TACO using Llama 3.1 70B - mean policy gradient achieves the best mean (pass@1) performance, while pass@$k$ gradient variants clearly achieve much better performance fo the pass@$k$ performance.

*Table 1.* The pass@$k$ performance of Llama 3.1 70B Instruct on CodeContests valid/test set up to pass@100. Models are trained on CodeContests training set. The pass@$k$ objectives optimize for $k = 8$.

| Method | Variant | CodeContests / Valid | | | CodeContests / Test | | |
|---|---|---|---|---|---|---|---|
| | | pass@1 | pass@10 | pass@100 | pass@1 | pass@10 | pass@100 |
| Policy Gradient | mean | **24.6** | 32.9 | 38.1 | **26.2** | 35.1 | 39.8 |
| | pass@k | 21.7 | **42.2** | **54.9** | 20.6 | **41.7** | 51.8 |
| | biased pass@k | 18.0 | 38.4 | 54.2 | 19.3 | 39.5 | **51.9** |
| PPO | mean | 20.0 | 27.5 | 33.5 | **24.9** | 34.6 | 41.1 |
| | pass@k | 19.2 | 37.3 | 52.4 | 21.2 | 41.4 | 51.0 |
| | biased pass@k | **21.6** | **41.7** | **55.8** | 22.6 | **45.1** | **56.3** |

**Integration with PPO.** We also investigate how the $k$-sample gradient variants can be combined with much more sophisticated algorithmic stack to illustrate its generic practical utility. Note that the pass@$k$ objective can be seamlessly integrated into other policy gradient variants, such as PPO (Schulman et al., 2017), which adds clipping and an additional value function that serves as the baseline. Figure 6 shows the performance of using PPO. We can observe that PPO with pass@1 objective enters in high-KL regime at the end of the training, due to the fact that its update generally leverages much more dense signals. The sparse nature of pass@$k$ objective keeps the policy under low-KL regime and maintains higher pass@8 performance. For the purpose of training language models with fixation on the reference policy (Ziegler et al., 2019; Ouyang et al., 2022), pass@$k$ objectives achieve a balance for the performance metrics.

**Model size.** We also include the experiment result with Llama 3.1 8B Instruct model in Appendix A.4. Compared to 70B, the performance differences across training objectives are less pronounced. We posit that the model size and the benchmark play a crucial role; for 8B model, the average pass@1 on CodeContests is modest, resulting in sparse reward signal. Intuitively, pass@$k$ objective will zero out the advantage for problems solved more than once. Therefore, we expect the difference to widen as the dataset contains more problems that the model is able to solve multiple times.

### 6.1. Generalization to different $k$ for evaluation

We investigate the impact of different training objectives, when combined with increased inference time budget. Concretely, we study how pass@$k$ performance scales with the number of samples $k$ at evaluation time.

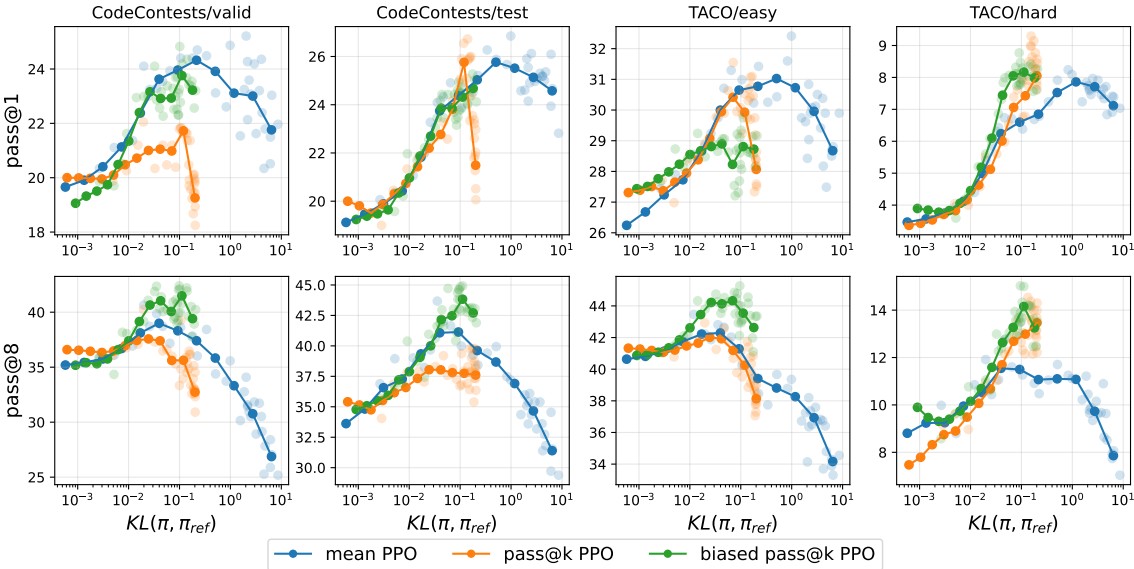

*Figure 6.* PPO code generation evaluation results with Llama 3.1 70B. We combine $k$-sample gradient estimates with a PPO implementation stack and compare with the default mean policy gradient baseline. We see that biased pass@$k$ gradient estimate especially achieves a good performance trade-off. Details of how the $k$-sample gradient variants are combined with PPO can be found in Appendix A.4.

*Table 2.* The pass@$k$ performance of Llama 3.1 70B Instruct on TACO *easy* and *hard* split up to pass@100. Models are trained on CodeContests training set. The pass@$k$ objectives optimize for $k = 8$. When integrated with pass@$k$ and biased pass@$k$ updates, both policy gradients and PPO have improved pass@$k$ performance, especially for large values of test time $k$.

| Method | Variant | TACO/easy | | | TACO/hard | | |
|---|---|---|---|---|---|---|---|
| | | pass@1 | pass@10 | pass@100 | pass@1 | pass@10 | pass@100 |
| Policy Gradient | mean | **29.9** | 34.5 | 38.2 | **8.1** | 9.9 | 11.8 |
| | pass@k | 29.6 | **43.2** | 48.9 | 7.8 | **14.4** | 18.0 |
| | biased pass@k | 26.7 | 42.5 | **51.5** | 7.0 | 14.3 | **21.9** |
| PPO | mean | 27.5 | 35.3 | 39.5 | 7.3 | 8.8 | 11.1 |
| | pass@k | **28.0** | 41.6 | 49.5 | 7.6 | 14.2 | 18.7 |
| | biased pass@k | 27.2 | **44.8** | **53.8** | **7.7** | **14.6** | 21.3 |

We show in Table 1 and Table 2 the pass@$k$ performance across different $k$ of Llama 3.1 70B Instruct model trained using different objectives. Models are trained with 6400 gradient update steps. For pass@$k$ objective, models trained to optimize pass@$k$ ($k = 8$) can generalize to pass rate with different $k$ up to $k = 100$. Amazingly, the performance improvements scales with $k$: for both CodeContest and TACO, the improvements go from $5 - 10\%$ at $k = 10$ to $10 - 20\%$ at $k = 100$. In general, improvements on hard and test splits are a bit less than improvements on easy and validation splits.

**Generalization to $k = 100$ at test time.** We show in Table 1 and Table 2 (in the Appendix) that Llama 3.1 70B Instruct model trained to optimize pass@$k$ ($k = 8$) can gen-

eralize to pass rate with different $k$ up to $k = 100$, with over $+10\%$ solve rate improvement over the baseline. on validation and test set.

# 7. Discussion and limitations

Lots of efforts remain to adapt the training methods to inference time objectives beyond this work. Applying the methodology here naively would incur large computational burden (Silver et al., 2016; 2018; Brown & Sandholm, 2018; 2019). Furthermore, many recent inference time algorithms do not treat each sample equally, such as self-correction, reflection and more complicated logic (Yao et al., 2022; Huang et al., 2023; Asai et al., 2023). This requires careful algorithmic improvements for better credit assignment.

## Impact Statement

This paper presents work whose goal is to advance fundamental algorithmic development. There are many potential societal consequences of our work, none which we feel must be specifically highlighted here.

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

# APPENDICES: Optimizing Language Models for Inference Time Objectives using Reinforcement Learning

## A. Hyper-parameters and experimental details

We experimented with the Llama 3 model of size 3B, 8B and 70B (Dubey et al., 2024). All experiments are conducted with identical hyper-parameter settings: we always apply a batch size of $B = 64$ prompts per update, and sample $k = 4$ distinct generations per prompt by default. All training and evaluation sampling are conducted at a temperature of $\tau = 1$ and with top-p $= 1$.

We train on the MATH training set with 7500 examples and evaluate on the test set with 5000 examples (Hendrycks et al., 2021). A supervised fine-tuning on the training set is conducted to warm up the RL training. As part of the ablation, we also train on the HARP dataset, which consists of about 4300 examples of difficult math problems harvested from a few public sources (Yue et al., 2024).

For both training and evaluation, we provide system instructions that ask the model to generate a response with step-by-step solution, followed by a final conclusion phrased as *the final answer is* followed by the answer predicted by the model. This is consistent with the prompt structure discussed for Llama models (Dubey et al., 2024; Yue et al., 2024).

### A.1. Ablation on number of samples $k$

**Pass@$k$.** Figure 7 shows the full results for the ablation results. We see as $k$ increases, the $k$-sample gradient estimates produce performance improvements on the pass@$k$ objective. More significant improvements are observed for the more difficult split of the dataset.

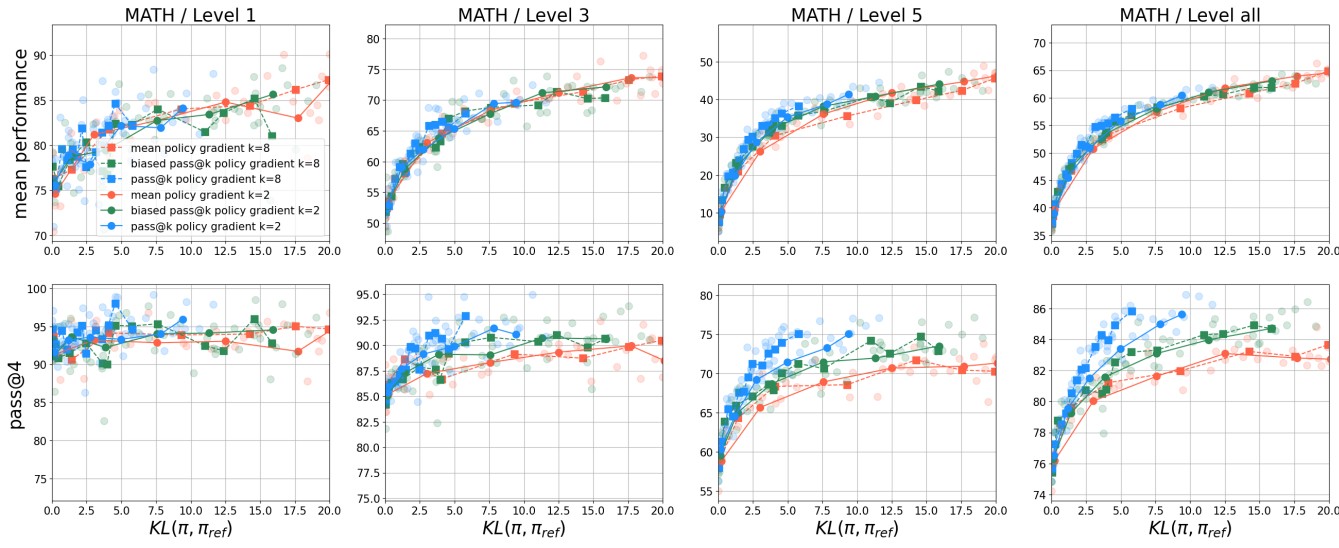

*Figure 7.* Ablation with number of samples $k$ for majority voting. We vary the number of samples $k$ for each gradient update for the pass@$k$ objective. We observe a more efficiency gains for the pass@$k$ gradient estimates compared to the policy gradient baseline. Importantly, note that as $k$ varies, the pass@$k$ algorithm changes its objectives.

**Majority voting.** Figure 8 shows the ablation results for the majority voting based results and $k$-sample gradient based algorithm. We observe a slight improvement of the $k$-sample based algorithms over policy gradient baseline, though the impact of the number of samples $k$ is less signficant compared to the pass@$k$ case.

### A.2. Training on HARP

We train both 8B and 70B on the HARP dataset, with both the pass@$k$ and majority voting based algorithms and objectives.

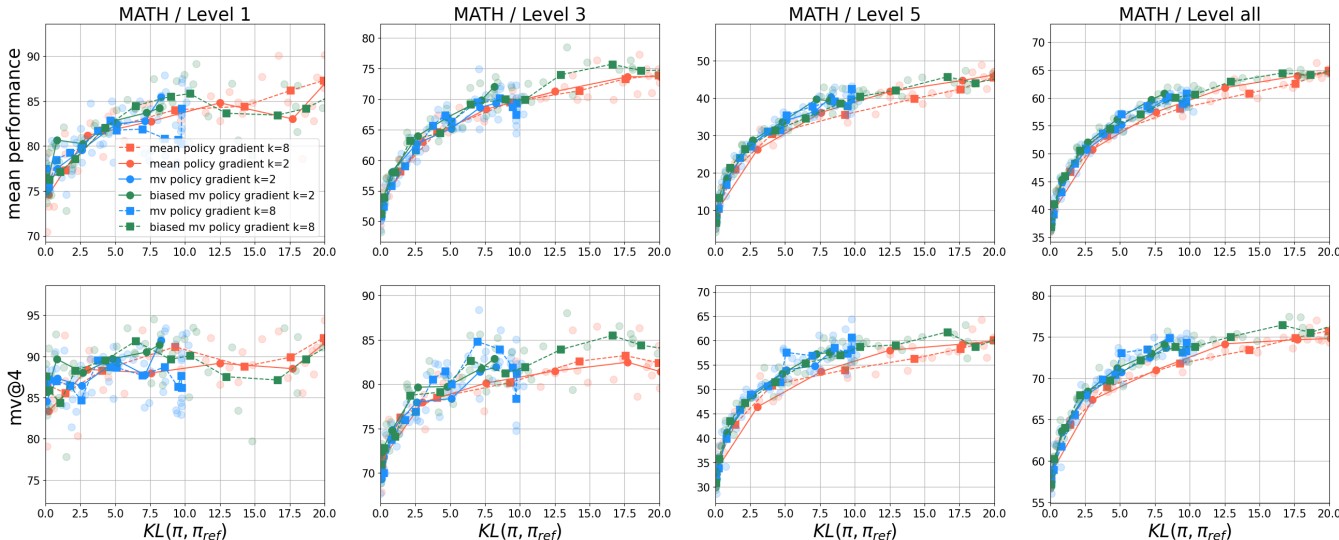

*Figure 8.* Ablation with number of samples $k$ for majority voting. We vary the number of samples $k$ for each gradient update for the pass@$k$ objective. We observe a modest efficiency gains for the pass@$k$ gradient estimates compared to the policy gradient baseline but the impact of the number samples $k$ is less significant.

**8B model.** Figure 9 and Figure 10 show the results for the pass@$k$ and majority voting respectively. Overall, we see that the regular policy gradient algorithm is modestly outperformed by the $k$-sample based algorithms, in terms of the KL-efficiency trade-off against the training performance.

However, one observation that is compatible with prior results, is that the regular policy gradient algorithm tends to allow the policy to deviate more from the reference policy using a fixed number of updates compared to the $k$-sample based algorithms. This presents a practical trade-off - if we want to obtain a good level of performance with a *fixed* number of training steps, the policy gradient algorithm might be a better option since it is more compute-efficient.

**70B model.** Figure 11 shows the HARP training results for the majority voting case. We see that the improvement is less clear for both the mean and the majority voting performance - the three algorithmic variants seem to obtain similar performance, with the $k$-sample policy gradient algorithm obtaining a very slight advantage over others.

### A.3. Evaluation

Figure 16 shows the evaluation performance for the majority voting on MATH, using the 8B model. Across the various baselines we compare, we note that the biased $k$-sample policy gradient algorithm seems to slightly outperform the policy gradient algorithm as KL is large, though the policy gradient algorithm itself is clearly a strong baseline.

The unbiased $k$-sample policy gradient algorithm seems to slightly underperform on evaluation, which contrasts the strong training performance in Figure 15. We speculate that this suggests an intriguing interaction between the training and testing performance worthy of further investigation: likely the merits of such $k$-sample based algorithms depend on the nature and the size of the dataset, which is better tested out in practical applications.

### A.4. Experimental details and additional result on CodeContests and TACO

**Experimental details.** We experimented with Llama 3.1 8B and 70B Instruct model. For both model we use the same hyperparameters. We use a learning rate $2e^{-7}$, constant learning rate scheduling with 50 warmup steps and weight decay of 0.1. We sample $k = 8$ generations per prompt. We update the model with a mini batch size 2 with sequence length 8192 and train in total 8k gradient update steps. In training, sampling is conducted at a temperature of $\tau = 1$ and with top-p $= 1$. In evaluation, we sample at a temperature of $\tau = 1$ and with top-p $= 0.95$. Both pass@1 and pass@8 are estimated out of 20 samples. We evaluate the correctness of generated Python code using the official codebase of Li et al. (2022). Our

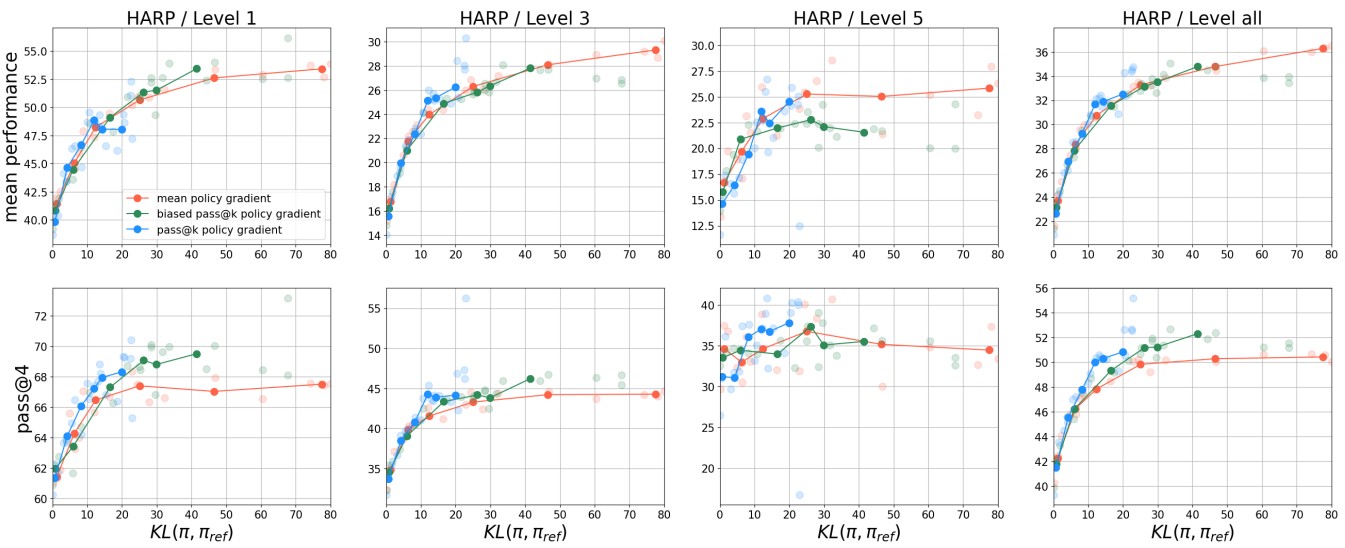

*Figure 9.* Training reward for pass@$k$ on the HARP dataset for 8B model. We can see observe relative performance efficiency of $k$-sample gradient variants compared to the baseline method. The gains are overall less significant compared to the 70B model, whose model capacity is more suitable for the challenging HARP dataset.

asynchronous online training decouples the training and the inference, for which we include an importance sampling term to correct the slightly off-policy nature between current policy and the behavior policy. The original CodeContests training set contains 13328 problems. We follow the decontamination process presented by Zheng et al. (2024) to decontaminate CodeContests training set against TACO evaluation set. We further remove problem instances with less than 5 test cases. This results in total 12275 problems which we use to train our model.

**Integrating with PPO.** We can integrate advantage clipping and an additional value model $V_\psi$ to serve as an additional baseline. Given a problem $x$, we sample multiple responses $\{y_1, y_2, ..., y_k\}$ from $\pi_{\theta_{old}}$, the training objective aims to maximize the following objective:

$$J^\pi(\theta) = \hat{\mathbb{E}}_{y \sim \pi_{\theta_{old}}} \left[ \min \left( \frac{\pi_\theta(y|x)}{\pi_{\theta_{old}}(y|x)} \hat{A}, \text{clip} \left( \frac{\pi_\theta(y|x)}{\pi_{\theta_{old}}(y|x)}, 1-\epsilon, 1+\epsilon \right) \hat{A} \right) \right]$$

$$= \frac{1}{k} \sum_{i=1}^{k} \min \left( \frac{\pi_\theta(y_i|x)}{\pi_{\theta_{old}}(y_i|x)} \hat{A}_i, \text{clip} \left( \frac{\pi_\theta(y_i|x)}{\pi_{\theta_{old}}(y_i|x)}, 1-\epsilon, 1+\epsilon \right) \hat{A}_i \right),$$

$$\hat{A}_i = R_i - V_\psi(x),$$

where $R_i$ takes the following different forms according to the training objective:

$$\text{Mean(pass@1) training objective:} \quad R_i = r_i$$

$$\text{Pass@}k \text{ training objective:} \quad R_i = \max_{j \in \{1,...,k\}} r_j - \max_{j \neq i} r_j$$

$$\text{Biased pass@}k \text{ training objective:} \quad R_i = \max_{j \in \{1,...,k\}} r_j - \max_{j \neq i} r_j - \frac{1}{k} \sum_{k=1}^{k} \left[ \max_{j \in \{1,...,k\}} r_j - \max_{j \neq k} r_j \right]$$

with $r_i$ being the $+1/-1$ binary reward of whether the code $y_i$ is correct. We train the value model $V_\psi$ to minimize the following clipped value loss:

$$L^V(\psi) = \hat{\mathbb{E}} \left[ \frac{1}{2} \max \left( (V_\psi(x) - R_i)^2, (\text{clip}(V_\psi(x), V_{\psi_{old}}(x) - \alpha, V_{\psi_{old}}(x) + \alpha) - R_i)^2 \right) \right].$$

We set $\epsilon = \alpha = 0.2$ in our experiments.

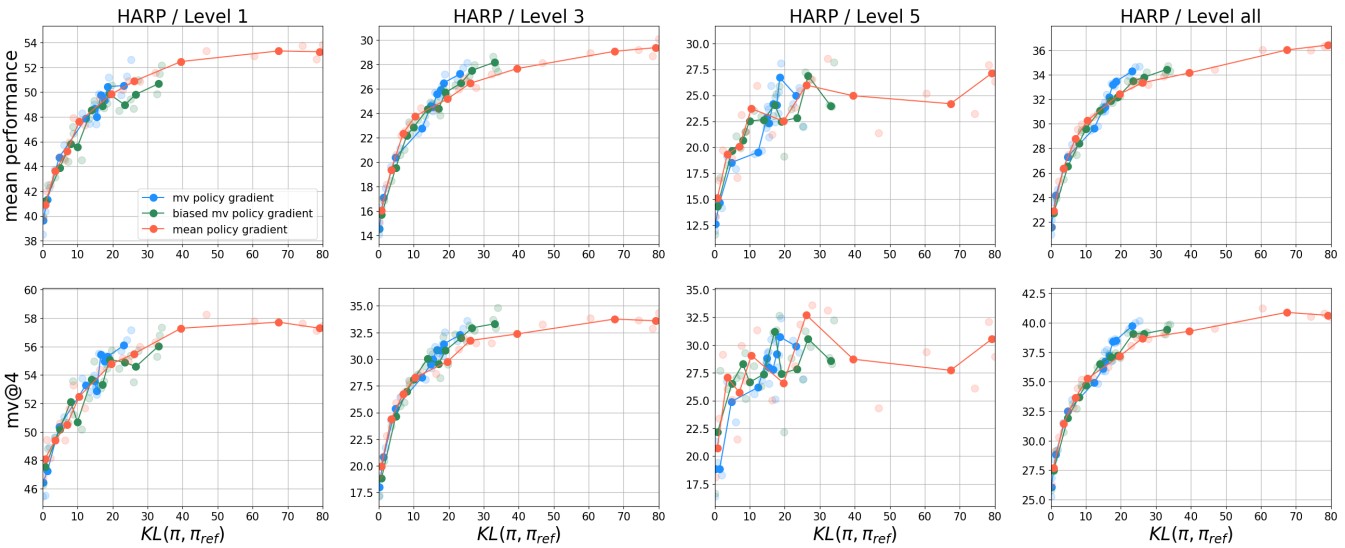

*Figure 10.* Training reward for majority voting on the HARP dataset for 8B model. Overall, the training performance improvement is less significant compared to the pass@$k$ case.

**Llama 3.1 8B Instruct Performance.** For Llama 3.1 8B Instruct model, we show in Figure 12 the performance of using mean policy gradient, pass@$k$ policy gradient and biased pass@$k$ policy gradient. We also show in Figure 13 the performance when integrating the training objective with PPO.

## B. Additional discussion

### B.1. Biased objective for majority voting

Following the recipe to derive biased $k$-sample objective for the $k$-sample objective, we discuss the case for majority voting. Assume there are $m$ unique answers and they are ordered by count $a_{(i)}, i \in [m]$. It is not difficult to show that, the resulting objective is a weighted objective of the most common and second most common answer

$$\mathbb{E}\left[P(y) \cdot r\left(a_{(m)}\right) + (1 - P(y)) \cdot r\left(a_{(m-1)}\right)\right],$$

where $P(y)$ is the probability that the leave-one-out majority voted answer is the second most common answer. Intuitively, the leave-one-out majority voted answer can either be $a_{(m)}$ (when for example $|a_{(m)}| \geq |a_{(m-1)}| + 2$) or the second most common answer $a_{(m-1)}$.

### B.2. Interpolation between $k$-sample objectives and mean objective

We have hinted at the observation that the biased $k$-sample objective is a smoother objective than the original $k$-sample objective. The biased objective also approaches the mean objective $\frac{1}{k} \sum_{i=1}^{k} r_i$ which is arguably the most smooth objective one can construct.

**Leave-$p$-out objectives.** Extending the idea of leave-one-out further, we discuss *leave-p-out*, which depicts a spectrum of objectives interpolating the original $k$-sample objective and the mean objective. We focus on the pass@$k$ objective as follows

$$\frac{1}{\binom{k}{p}} \mathbb{E}\left[\sum_{s \in S_p} \max_{i \notin s} r_i\right], \tag{5}$$

where $S_p$ is the set of all subsets of $p$ indices from $\{1, \ldots, k\}$, with $\binom{k}{p}$ being its cardinality. Notice that for $p = 0$ we recover the initial problem of the pass@$k$ objective. For $p = 1$ we get the de-meaned objective introduced in the previous

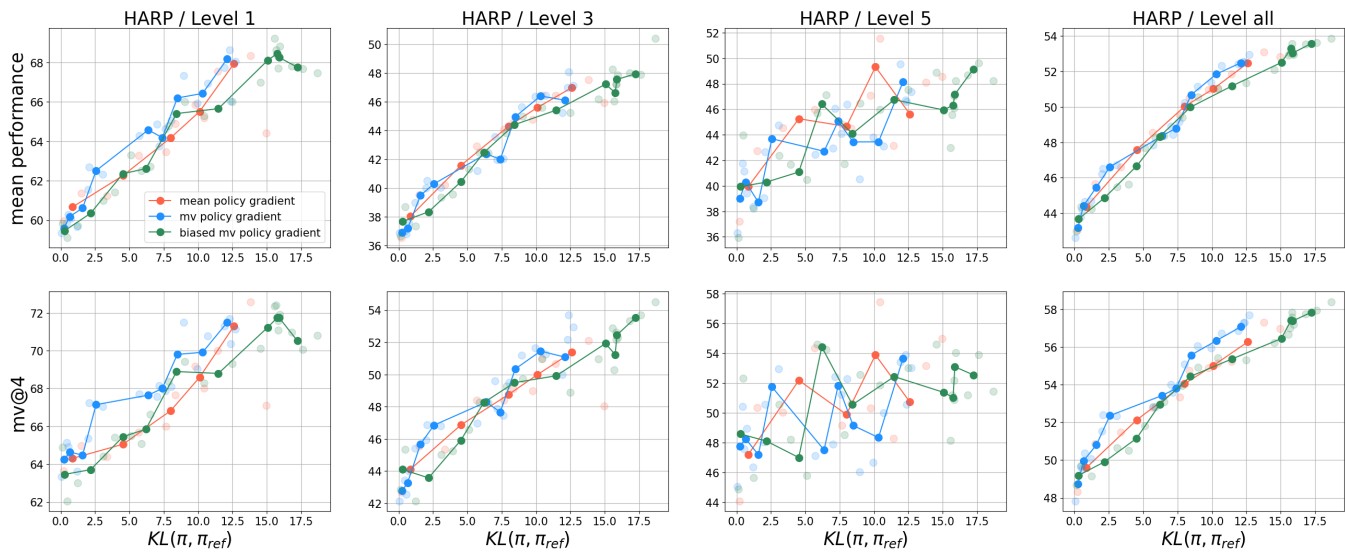

*Figure 11.* Training reward for majority voting on the HARP dataset for 70B model.

section. For $p = k - 1$ we get the problem of maximizing the expected rewards. For $p = 2$ we get the objective

$$\frac{2}{k(k-1)} \mathbb{E} \left[ \sum_{i \neq j} \max_{l \neq \{i,j\}} r_l \right],$$

A practical implementation of a gradient estimate of the objective (5) can be written as the de-meaned gradient using the advantages $\max_i r_i - \max_{i \notin s} r_i$ for $s \in S_p$. Indeed:

$$\nabla \mathbb{E} \left[ \frac{1}{\binom{k}{p}} \sum_{s \in S_p} \max_{i \notin s} r_i \right]$$

$$= \mathbb{E} \left[ \left( \sum_{i \in \{1,\ldots,k\}} \nabla \log \pi(y_i) \right) \frac{1}{\binom{k}{p}} \sum_{s' \in S_p} \max_{i \notin s'} r_i \right]$$

$$= \mathbb{E} \left[ \frac{1}{\binom{k-1}{p-1}} \sum_{s \in S_p} \left( \sum_{i \in s} \nabla \log \pi(y_i) \right) \left( \frac{1}{\binom{k}{p}} \sum_{s' \in S_p} \max_{i \notin s'} r_i \right) \right]$$

$$= \frac{1}{\binom{k-1}{p-1}} \mathbb{E} \left[ \sum_{s \in S_p} \left( \sum_{i \in s} \nabla \log \pi(y_i) \right) \left( \frac{1}{\binom{k}{p}} \sum_{s' \in S_p} \max_{i \notin s'} r_i - \max_{i \notin s} r_i \right) \right]$$

$$= \frac{1}{\binom{k-1}{p-1}} \mathbb{E}_\pi \left[ \sum_{s \in S_p} \left( \sum_{i \in s} \nabla \log \pi(y_i) \right) \left( \underbrace{\max_i r_i - \max_{i \notin s} r_i}_{=:A_s} - \frac{1}{\binom{k}{p}} \sum_{s' \in S_p} A_{s'} \right) \right].$$

For example, the corresponding gradient for $p = 2$ is

$$\frac{1}{k-1} \mathbb{E}_\pi \left[ \sum_{i \neq j} (\nabla \log \pi(y_i) + \nabla \log \pi(y_j)) \left( \underbrace{\max_l r_l - \max_{l \notin \{i,j\}} r_l}_{=:A_{i,j}} - \frac{2}{k(k-1)} \sum_{i' \neq j'} A_{i',j'} \right) \right].$$

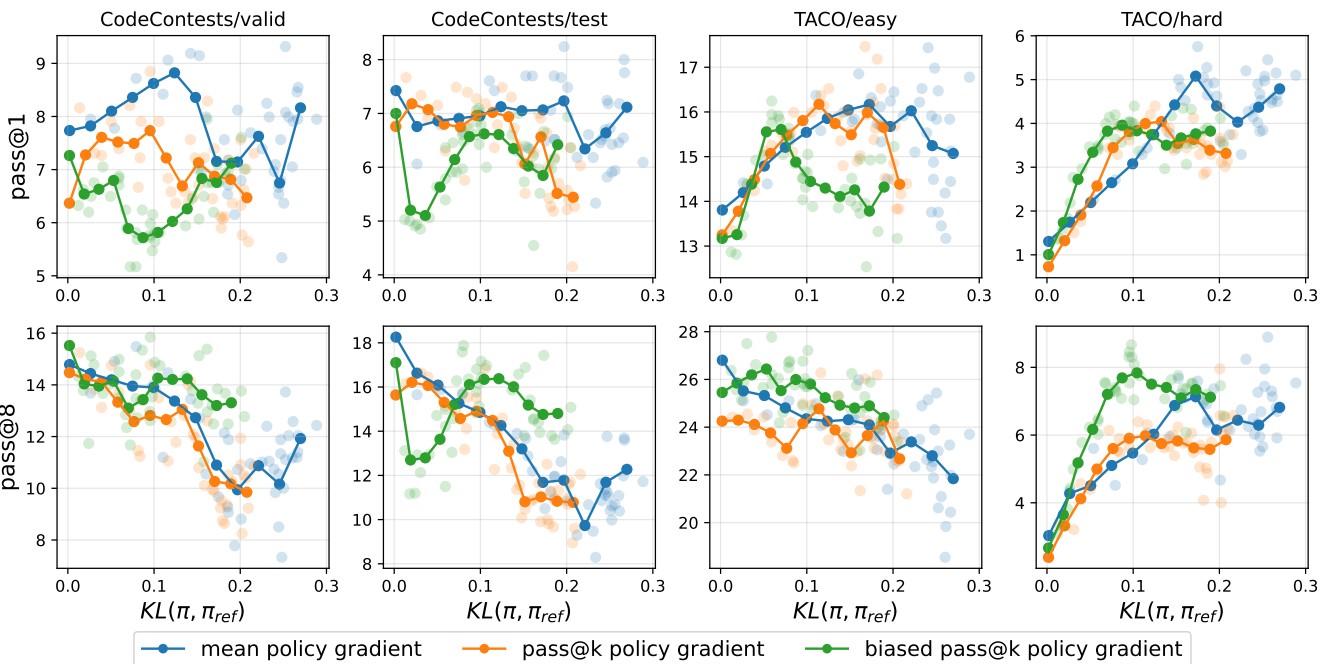

*Figure 12.* Llama 3.1 8B Instruct performance on CodeContests and TACO. We compare 3 methods: the mean policy gradient with leave-one-out control variate that optimizes for pass@1, the pass@$k$ policy gradient and the biased pass@$k$ policy gradient with mean advantage serving as the baseline.

---

**Algorithm 1** Online policy optimization

---

1: **INPUT** policy $\pi_\theta$
2: **while** $t = 0, 1, 2...$ **do**
3:   (i) Sample prompt $x \sim \rho$
4:   (ii) Collect $k$ trajectories per prompt $(y_i)_{i=1}^k$ from $\pi_\theta(\cdot|x)$ for all $i \in \{1, 2...k\}$.
5:   (iii) Update policy parameter $\theta$ based on one of the gradient variants above (Eqn (2) or Eqn (3)).
6: **end while**

---

**Softmax objectives.** An alternative approach to interpolate between $k$-sample objective and the mean is to through softmax. For example, consider the objective

$$\mathbb{E}\left[\sum_{i=1}^{k} p_i r_i\right]$$

where $p_i \propto \exp(\beta r_i)$ is the softmax distribution scaled by parameter $\beta \geq 0$. The above objective approaches mean when $\beta = 0$ and pass@$k$ when $\beta \to \infty$. We find that the softmax objective tends to produce much more learning signal than pass@$k$, since its advantage estimates are generally more dense.

In general, we might also make use of leave-one-out softmax objectives as baselines for variance reduction as opposed to leave-one-out pass@$k$. This makes sure that the optimization objective is intact while introducing generally smoother signals.

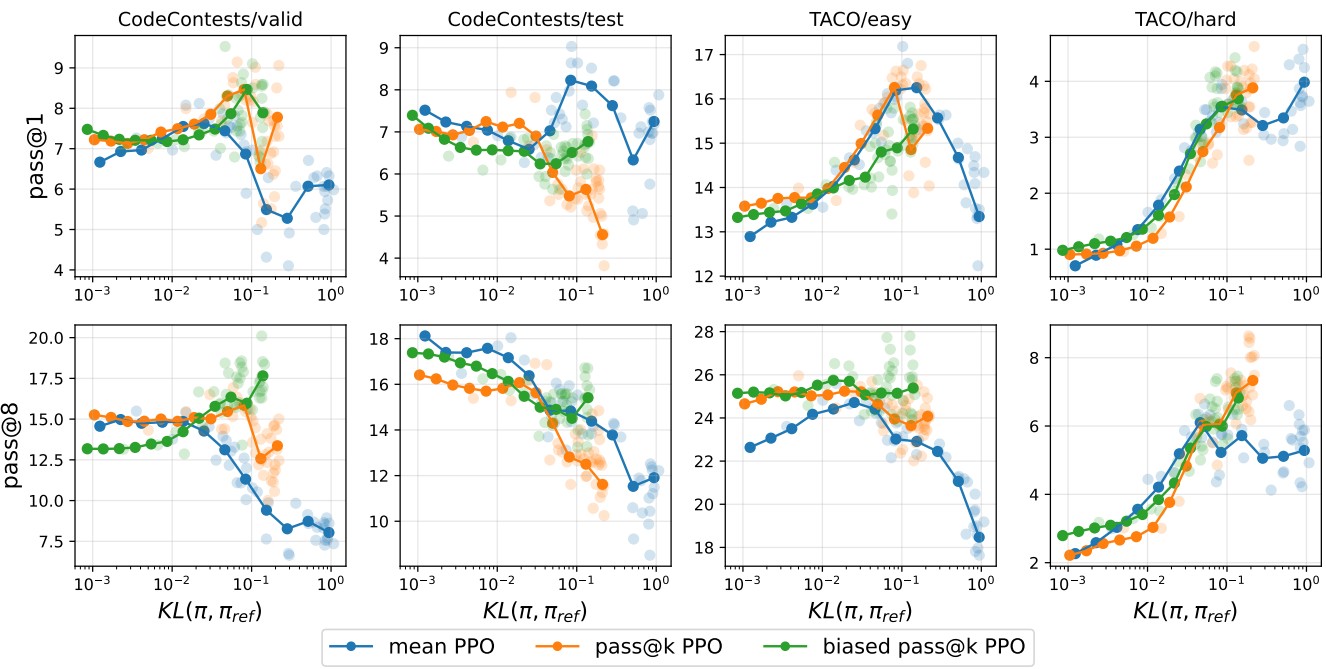

*Figure 13.* Llama 3.1 8B Instruct performance on CodeContests and TACO. We incorporate the PPO training objective and compare 3 variants: the vanilla PPO objective that optimizes the pass@1 performance, the pass@$k$ objective and the biased pass@$k$ objective with mean advantage serving as the baseline.

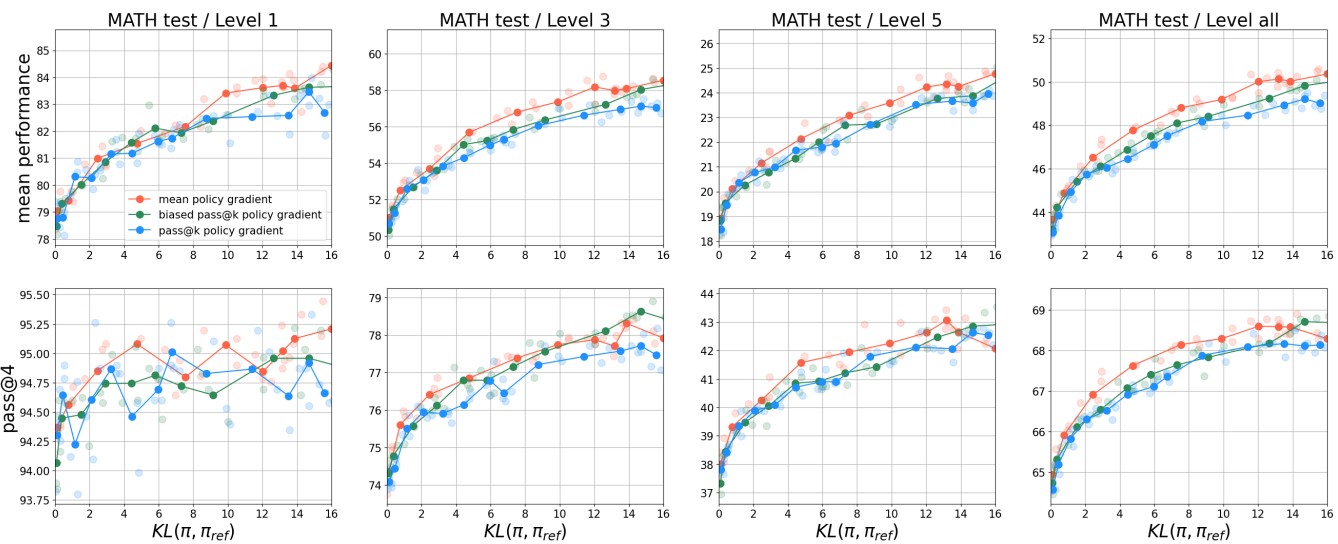

*Figure 14.* MATH test 8B model. We carry out evaluation on the MATH test set, using the same sampling hyper-parameter as the training time. We observe that the regular policy gradient baseline obtains a strong performance for pass@$k$ with $k \in \{1, 4\}$. However, as $k$ increases, it tends to be outperformed by the $k$-sample gradient variants largely due to a more significant drop in the sample diversity.

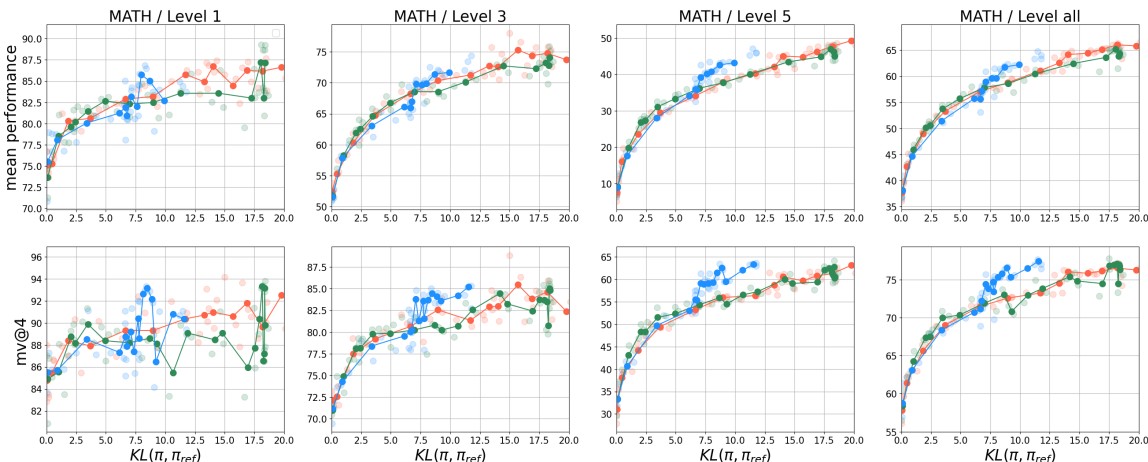

*Figure 15.* MATH training majority voting 8B model. We compare three baselines: regular mean policy gradient algorithm and two variants of majority voting policy gradient algorithms (unbiased and biased). We observe the unbiased majority voting policy gradient algorithm improves slightly over the other two alternatives.

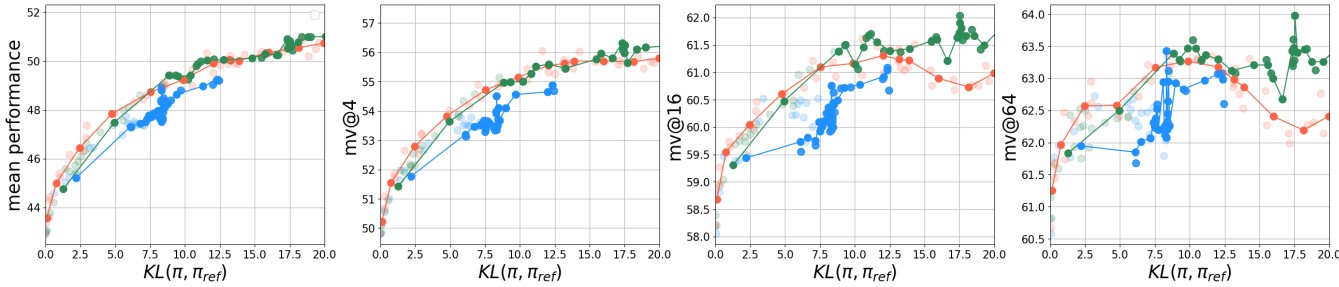

*Figure 16.* MATH test 8B model for majority voting. We carry out evaluation on the MATH test set, using the same sampling hyper-parameter as the training time. We observe that the regular policy gradient baseline obtains a strong performance overall, though slightly outperformed by the biased $k$-sample gradient algorithm. The original unbiased $k$-sample gradient algorith, despite performing better at training reward, is slightly underperforming for evaluation.

