# OpenReview forum: "Optimizing Language Models for Inference Time Objectives using Reinforcement Learning"
_ICML.cc/2025/Conference — ICML 2025 poster_

### Official Review · Reviewer_bYLj · 2025-03-11

**Overall Recommendation:** 3

**Summary:**

This paper explores the benefits of explicitly training language models using reinforcement learning to perform well under inference time algorithmic techniques like pass@$k$ and majority voting. The authors argue that directly optimizing for inference time objectives can lead to improved performance on those specific metrics.
This paper provides a formulation for $k$-sample objectives and efficient gradient estimation techniques. The authors derive an unbiased gradient estimator using the leave-one-out method, and a biased variant that reduces variance by centering the advantage function.
The paper demonstrates these approaches on mathematical reasoning (MATH dataset) and code generation tasks (CodeContests), showing performance benefits when training directly for inference time objectives compared to training for mean performance.

**Claims And Evidence:**

Yes

**Essential References Not Discussed:**

No

**Experimental Designs Or Analyses:**

Yes

**Methods And Evaluation Criteria:**

Yes

**Other Comments Or Suggestions:**

### Comments

In general, I think this is a comprehensive paper, which tells a complete story about the gap between mean performance objective and pass@$k$ objective, showcasing the inference-time RL method better aligns with the latter. However, this result is not surprising, as stated in weaknesses.

### Suggestions

The authors should use the same colors in different figures. Currently Figure 2 and 3 have different colors for the same algorithms, which is highly confusing in the first sight.

**Other Strengths And Weaknesses:**

### Strengths

- **Technical Contribution:** The theoretical framework for optimizing $k$-sample objectives extends beyond previous work in multi-sample objectives. The proposed estimators are verified to be KL-efficient than directly optimizing mean performance on benchmarks.
- **Extensive Empirical Validation:** The authors conduct thorough ablation studies across model sizes (3B, 8B, 70B), different values of $k$, and various datasets, strengthening their claims.

### Weaknesses

- **Result Within Expectation:** Since there is gap between mean performance objective and pass@$k$ objective, the results in this paper is normally within expectation. It seems unfair to compare with mean policy gradient, which is not talored for pass@$k$ objective.
- **Computational Efficiency:** The $k$-sample gradient methods require $k$ times more inference computation during training, which might limit practical applicability for very large models, especially because generating a sample is the most time-consuming process.
- **Task-Dependent Performance:** As acknowledged by the authors, the benefits vary significantly based on task difficulty - for easier tasks, the standard policy gradient approach remains competitive (and more efficient in samples).

**Questions For Authors:**

Regarding the computational efficiency issue mentioned in weaknesses, can the authors provide another ablation study on the **total number of samples**? Namely, allow mean policy gradient to do $k$ times of update than the proposed algorithms, and plot the figures.

**Relation To Broader Scientific Literature:**

This paper makes a contribution to language model training by showing a difference between and inference objectives. Most prior work has treated these as separate concerns, but this approach directly incorporates inference techniques into the training process.
For practical applications like code generation, where pass@$k$ metrics are commonly used in evaluation, this approach provides a clear path to improving performance.

**Theoretical Claims:**

Yes

---

> ### Author Rebuttal · Authors · 2025-03-31
>
> We thank the reviewer for the time and efforts in providing us with the valuable reviews. We will incorporate your feedback into our later revision. We address your comments below.
>
>
> > Regarding the computational efficiency issue mentioned in weaknesses, can the authors provide another ablation study on the total number of samples? Namely, allow mean policy gradient to do k
>  times of update than the proposed algorithms, and plot the figures.
>
> In our manuscript the baseline (mean PG, and mean PPO), already uses the same number of samples as the other variants to enable a fair comparison in terms of compute cost.
>
> We also conducted exp. using k samples for using the PPO loss but remove the value model. For mean objective, this corresponds to the GRPO loss.
> For the pass@k objective, note that the proposed biased pass@k objective already subtracts the mean out of k samples, which coincides with the Dr.GRPO loss recently proposed https://arxiv.org/abs/2503.20783 (only subtract the mean but not divide by the std. deviation).
>
> For completeness, we also include:
> 1. the mean objective using Dr. GRPO loss
> 2. biased pass@k objective using GRPO loss, i.e., further divide by the std. Deviation
>
>
> Due to time and resource constraints we are only able to finish the exp. using llama 3.1 8b model on the coding tasks. We will add the others in the revision.
>
> Checkpoints are all trained for 8000 gradient steps using the same set of hyperparameters. We report the number by avg the perf of the last 5 ckpts, (8k, 7.8k, …, 7.2k) steps, to ensure a non cherry-picked results.
>
> pass@1:
>
> | Method                          | CodeContests/Valid | CodeContests/Test | TACO/easy | TACO/hard |
> |----------------------------------|--------------------|-------------------|-----------|-----------|
> | **Mean (Dr. GRPO)**              | 7.17               | 8.88              | 16.47     | 5.40      |
> | **Mean (GRPO)**                  | 9.21               | 8.39              | 17.89     | 4.74      |
> | **Biased pass@k (Dr. GRPO)**     | 6.79               | 7.03              | 15.87     | 4.86      |
> | **Biased pass@k (GRPO)**         | 8.11               | 6.67              | 14.51     | 4.12      |
>
> pass@10:
>
> | Method                          | CodeContests/Valid | CodeContests/Test | TACO/easy | TACO/hard |
> |----------------------------------|--------------------|-------------------|-----------|-----------|
> | **Mean (Dr. GRPO)**              | 10.61              | 14.89             | 22.91     | 6.89      |
> | **Mean (GRPO)**                  | 13.88              | 14.41             | 24.11     | 7.06      |
> | **Biased pass@k (Dr. GRPO)**     | 15.71              | 16.44             | 27.01     | 8.55      |
> | **Biased pass@k (GRPO)**         | 16.83              | 15.45             | 26.08     | 7.37      |

---

> > ### Comment · Reviewer_bYLj · 2025-04-03
> >
> > I appreciate the authors for their updated experiments. My concerns are addressed and I'll raise my score accordingly.

---

> > > ### Author Response · Authors · 2025-04-03
> > >
> > > Thank you very much for engaging with our rebuttal. We sincerely appreciate you adjusting the scores. Please also let us know if you have further questions.

---

### Official Review · Reviewer_FCtK · 2025-03-14

**Overall Recommendation:** 2

**Summary:**

This paper investigates the impact of reinforcement learning (RL) objectives that optimize multi-sample metrics such as Pass@k and Majority Voting (Maj@k). It contrasts these objectives with standard mean reward RL objectives across mathematical reasoning and code generation tasks. The study highlights the trade-off between optimizing for multi-sample inference objectives and maintaining strong Pass@1 performance.

**Claims And Evidence:**

Improved Pass@k Performance: The paper demonstrates that training with Pass@k policy gradient (PG) and biased Pass@k PG improves Pass@k performance. Supporting evidence is presented in Sections 4 and 5.
Effect of Increasing k During Training: The paper examines the impact of larger k values on performance (Section 4.2).

**Essential References Not Discussed:**

Variational Best-of-N Alignment (https://arxiv.org/abs/2407.06057v1)

**Experimental Designs Or Analyses:**

- The paper effectively demonstrates that improving Pass@k often comes at the expense of Pass@1 performance, particularly in code generation tasks.
Practical Limitations of Pass@k: Pass@k is not a practical metric in real-world inference scenarios as it requires an oracle verifier. While the authors argue that Pass@k represents an upper bound on inference-time scaling, it is unclear if this upper bound is useful given the lack of a reliable verifier (DeepSeek-AI, 2025).
- The field is shifting towards inference compute scaling within a single trajectory (e.g., longer chains of thought) rather than multi-sample methods. The paper does not address how its proposed techniques integrate with this paradigm.
- The authors compare their metrics against KL divergence from a reference policy but do not justify why this comparison is meaningful. If the primary objective is to improve Pass@1 and Pass@4, it is unclear why lower KL divergence is relevant.
- The primary MATH benchmark results are reported for the training set in the main paper, while test set results are relegated to the appendix. This should be reversed.
Baseline Comparisons: A simple PG with an entropy penalty should be included as a baseline for a fairer comparison.

**Methods And Evaluation Criteria:**

The paper evaluates three RL Objectives: Mean Policy Gradient, Pass@k Policy Gradient, Biased Pass@k Policy Gradient

Performance is primarily assessed through Pass@1 and Pass@4 scores while comparing these methods against KL with a reference policy. Experiments are conducted on three datasets:
MATH (Mathematical Reasoning), HARP (More difficult Mathematical Reasoning), CodeContests (Competitive Programming)

**Other Comments Or Suggestions:**

I don't have any other comments.

**Other Strengths And Weaknesses:**

Strengths:
I think the paper is well-organized and easy to follow. It evaluates reasoning and coding tasks across diverse datasets and honestly the analytical insights provide useful perspectives on the optimization of multi-sample objectives.

Weaknesses:
- I think the gains in Pass@k come at a significant cost to Pass@1, particularly in coding tasks. Importantly, Mean PG outperforms other multi-sample objectives on the MATH test set, even at Pass@4.
- The approach depends on having a strong verifier, raising doubts about practical improvements from these objectives.
- The paper lacks a comparison with standard PG + entropy bonus.
- The majority of the related work section is relegated to the appendix, omitting key citations from the main text.  Important prior work, such as Variational Best-of-N Alignment, is not discussed.

**Questions For Authors:**

I think I have discussed my questions in strengths and weaknesses section.

**Relation To Broader Scientific Literature:**

The paper focuses on the generator-verifier framework, where increased inference-time compute is allocated to generating multiple solutions and selecting the best one. However, recent trends have shifted toward single-trajectory inference-time scaling (e.g., longer chain-of-thought reasoning), limiting the practical applicability of methods discussed in the paper.

**Theoretical Claims:**

The paper provides analytical insights into optimizing inference-time objectives but does not introduce major theoretical results.

---

> ### Author Rebuttal · Authors · 2025-03-31
>
> We thank the reviewer for the time and efforts in providing us with the valuable reviews. We will incorporate your feedback into our later revision. We address your comments below.
>
> > I think the gains in Pass@k come at a significant cost to Pass@1, particularly in coding tasks.
>
> One major takeaway message is that there is a trade-off between pass@k and pass@1 in general. For coding tasks, pass@k is generally a more meaningful objective than pass@1 because we focus more on harder tasks, and obtaining one valid solution to a hard problem is enough [1].
>
> [1] Li et al, Competitive level code generation with AlphaCode, 2022
>
> >  Importantly, Mean PG outperforms other multi-sample objectives on the MATH test set, even at Pass@4.
>
> We are being truthful about the train/test generalization gap here - the performance improvements of our algorithm are better assessed at the training set, since the algorithm does not account for train/test generalization. The generalization gap might depend on multiple factors out of our control, such as the choice of the base model and data used for pretraining the models.
>
> It is true that the mean PG outperforms multi-sample objectives with MATH at pass@4, but the outperformance is marginal and limited to pass@4. The overall conclusions about the performance improvement from multi-sample gradient estimate still stand, but we are truthful about the fact that it does not necessarily generalize in all cases.
>
> > The approach depends on having a strong verifier, raising doubts about practical improvements from these objectives.
>
> This is not true. We do not require verifiers at all during training, unlike Chow et al. Can the reviewer clarify on the claim that the approach depends on a strong verifier and where it is used?
>
> For both training and evaluations, we use the standard numpy package to compute the rewards for training. This is common practice in the reasoning LLM literature.
>
> > The paper lacks a comparison with standard PG + entropy bonus.
> The entropy bonus in PG algorithms is complementary to the approach we take. In other words, in principle, our method can be combined with an entropy bonus as-is.
>
> As a result, we do not think adding entropy bonus comparison offers new insights into the current algorithmic designs. Entropy bonus might lead to performance improvements as a result of avoiding premature collapse, but it also introduces a tunable hyper-parameter, whereas our approach is hyper-parameter free compared to the baseline.
>
> > The majority of the related work section is relegated to the appendix, omitting key citations from the main text. Important prior work, such as Variational Best-of-N Alignment, is not discussed.
>
> We apologize for the inconvenience of moving most related work discussion to the Appendix due to ICML page limit. When the limit is relieved (camera ready), we can move much of the discussion into the main paper.
>
> Variational BoN: this is a related work and we will discuss it. Variational BoN does not optimize for the original objective as-is, but rather requires a variational approximation. This leads to an algorithm akin to the one introduced in Balashankar et al, and in a pairwise comparison setting. In comparison, we focus on the point-wise reward case and require no approximation to the original objective of interest.

---

> > ### Comment · Reviewer_FCtK · 2025-04-09
> >
> > Thank you for your effort and detailed answer. I have read your rebuttal. I think my main issues still stand. Regarding the verifier: A high pass@K can be useful when we can sample multiple times and then choose the correct answer between the K times. If we cannot do that in test time, it is not clear to me how we can use this LLM with high pass@K.

---

> > > ### Author Response · Authors · 2025-04-09
> > >
> > > Thank you very much for engaging with our rebuttal, we appreciate your time.
> > >
> > > > Regarding the verifier: A high pass@K can be useful when we can sample multiple times and then choose the correct answer between the K times. If we cannot do that in test time, it is not clear to me how we can use this LLM with high pass@K.
> > >
> > > We agree with the usefulness of pass@k with a high value of k. There is no indication in the paper that we cannot use a high value of k at test time.
> > >
> > > In fact, in Table 1 we show this situation: at training time we might train with a smaller value of k=8 while testing it for k=10,100. The performance does generalize and transfer to higher value of k. Is this what you have in mind?
> > >
> > > Please let us know if you have further questions on our paper. If our response addresses your technical points, please consider adjusting our scores too.

---

### Official Review · Reviewer_xLpG · 2025-03-16

**Overall Recommendation:** 2

**Summary:**

This paper explores the potential benefits of explicitly training LLMs for test-time inference. The authors introduce a new RL objective, which explicitly utilizes multiple samples and optimizes LLMs for inference-time objectives, like better pass@k performance or better majority voting performance. The empirical experiments show improvements on various math reasoning tasks and code generation tasks compared with baseline algorithms.

**Claims And Evidence:**

See “Methods And Evaluation Criter” and “Experimental Designs Or Analyses”.

**Essential References Not Discussed:**

The key contribution is a multi-sample RL objective that aligns with test-time inference. However, single-sample RL algorithm work is not well-discussed in this paper, such as [1], [2], and [3].

[1] Kumar, Aviral, et al. "Training language models to self-correct via reinforcement learning." arXiv preprint arXiv:2409.12917 (2024).

[2] Kazemnejad, Amirhossein, et al. "Vineppo: Unlocking rl potential for llm reasoning through refined credit assignment." arXiv preprint arXiv:2410.01679 (2024).

[3] Shao, Zhihong, et al. "Deepseekmath: Pushing the limits of mathematical reasoning in open language models." arXiv preprint arXiv:2402.03300 (2024).

**Experimental Designs Or Analyses:**

As mentioned in the Methods and Evaluation Criteria section, on one hand, I don't think it is a valid assumption to assume which test-time algorithm to use during training. On the other hand, for general RL algorithms, I suspect the performance of algorithms like GRPO, that utilize the same number of samples, will even outperform the objective proposed in this paper. To be clear, GRPO [1] doesn't require any prior assumption on test-time algorithm, but we can still evaluate it on both approaches. As shown in Figure 18 and Figure 19 from [2], the R1-Distill-Qwen-7B model, a single model, outperforms both maj@k and pass@k separately-trained 8B models reported in this paper on MATH. This is not a fair comparison since R1-Distill-Qwen-7B is a stronger base model, but my argument is mainly to question the necessity of separate objectives; a more rigorous comparison is needed. Baselines like GRPO and other single-objective functions with the same sample complexity are necessary for comparison.

[1] Shao, Zhihong, et al. "Deepseekmath: Pushing the limits of mathematical reasoning in open language models." arXiv preprint arXiv:2402.03300 (2024).

[2] Qu, Yuxiao, et al. "Optimizing Test-Time Compute via Meta Reinforcement Fine-Tuning." arXiv preprint arXiv:2503.07572 (2025).

**Methods And Evaluation Criteria:**

I am not fully convinced by the motivation of this work. Model training doesn't explicitly account for the downstream inference time algorithm. As presented in work [1], there is no single inference-time algorithm that works best for any scenario, and usually, during model training, it is not practical to assume what inference-time algorithm will be used. It's a user decision, determined by computation budget, latency requirements, etc. However, the objective proposed in this work, if I understand correctly, seems to require separate training for different objectives, which doesn't seem to be well-justified.

[1] Gao, Peizhong, et al. "Meta reasoning for large language models." arXiv preprint arXiv:2406.11698 (2024).

**Other Comments Or Suggestions:**

See “Methods And Evaluation Criter” and “Experimental Designs Or Analyses”.

**After Reply Rebuttal Comment**

I maintain my concern about the generalizability of the method. As the authors mentioned in "Reply Rebuttal Comment":

> If we train with pass@k and evaluate with maj@k, there is generally no guarantee that we can do better than pass@1.

I think it's a strong assumption that "inference time objective is known to inform the training time procedure." For multiple objectives during inference time, the computational overhead of retraining the model to optimize each target is concerning, and achieving better results by specifically optimizing for each objective seems less appealing. However, the authors have addressed my other concerns; therefore, I have updated my score accordingly.

**Other Strengths And Weaknesses:**

See “Methods And Evaluation Criter” and “Experimental Designs Or Analyses”.

**Questions For Authors:**

There are no implementation details in the paper. How are the hyperparameters selected for different settings in the paper?

**Relation To Broader Scientific Literature:**

This work is related to the claim that improving inference time performance can exceed the benefits of additional training time.

**Theoretical Claims:**

The proof in section 3 is correct.

---

> ### Author Rebuttal · Authors · 2025-03-31
>
> We thank the reviewer for the time and efforts in providing us with the valuable reviews. We will incorporate your feedback into our later revision. We address your comments below.
>
> > On the other hand, for general RL algorithms, I suspect the performance of algorithms like GRPO, that utilize the same number of samples, will even outperform the objective proposed in this paper.
>
> We strongly disagree with the reviewer.
>
> In our manuscript the baseline (mean PG, and mean PPO), as what the reviewer refers to as “general RL algorithm”, already uses the same number of samples as the other variants to enable a fair comparison. And we already provide empirical evidence in our paper that they do not outperform our proposed variants that optimize for pass@k.
>
> In other words, general RL algorithms such as GRPO only optimize for the mean performance, and would similarly benefit from algorithmic improvements proposed in this work. GRPO is not designed to optimize for pass@k on its own.
>
> > Baselines like GRPO and other single-objective functions with the same sample complexity are necessary for comparison.
>
> Please refer to our response to Reviewer bYLj for exp. on GRPO; we also include Dr. GRPO as a variant. Note that, our method could be integrated with various policy gradient variants, e.g., the PPO or vanilla Policy Gradient in the manuscript. Please refer to our response to Reviewer 61s6 for results of SFT and Rejection Sampling Finetuning (RFT) for additional comparison.
>
> > To be clear, GRPO [1] doesn't require any prior assumption on the test-time algorithm, but we can still evaluate it on both approaches.
>
> GRPO (or any policy gradient method derived from the objective $\max E(f(y)))$ **does** have prior assumptions on the test-time algorithm: the one that uses the exact protocol as the $E(f(y))$, i.e., pass@1. The reviewer refers to [2] to argue that a model trained using $\max E(f(y))$ objective works well on pass@k metrics, i.e., $\max E(f(y_1, … y_k))$ objective. This could be also true on the other way round.
>
> We motivate this from the first principle that such an objective is to maximize the k-sample objective $E(f(y_1, … y_k))$, contrary to the common formulation of maximizing cumulative return $E(f(y))$, where the return is a function for individual sample. Our proposed scheme works for settings that fit into this objective $E(f(y_1, … y_k))$. Specifically, we show that in the case of $f = r \circ \text{maj}$ and $f = \max$, this corresponds to the maj@k and pass@k metrics in math reasoning and code generation and gives empirical evidence.
>
> That said, our method also can be evaluated in multiple sampling budgets in test time (we do not set a constraint that training on k = 8 only works for k = 8 in test time, see our section of generalization to k = 100).
>
> Also, we share exactly the same hyperparameter as GRPO: the number of samples per prompt.
>
> > As shown in Figure 18 and Figure 19 from [2], the R1-Distill-Qwen-7B model, a single model, outperforms both maj@k and pass@k separately-trained 8B models reported in this paper on MATH. This is not a fair comparison since R1-Distill-Qwen-7B is a stronger base model, but my argument is mainly to question the necessity of separate objectives.
>
> The reviewer suggests that the access to a stronger model R1-Distill-Qwen-7B questions the necessity of separate training objectives.
>
> We object to this argument because the R1-Distill-Qwen-7B model is distilled from a capable mode R1, which is specifically trained via RL methods, which is the focus of our paper.
>
> This does not undermine the necessity of separate objectives but rather reinforces the validity and potential benefits of explicitly targeting inference-time objectives during training, which results in a better model in terms of pass@k performance and could bring the pass@k superiority to other models by following the same distillation process. Also, Figure 18 and Figure 19 from [2] is reported on MATH500, while our numbers are on a full MATH test set (a total 2500 problems).
>
> > There are no implementation details in the paper. How are the hyperparameters selected for different settings in the paper?
>
> Please find implementation details in the Appendix A (in particular A. and A.4).
>
> For sampling, we use top-p=1 and temperature $\tau=1$ sampling with standard parameters, as they are compatible with the RL training, though evaluations can be executed with various sampling scheme.
>
> For training, the experiments are rather robust to the choice of learning rate and other training parameters: Therefore, we fixed these hyperparameters among different settings (mean/pass@k objective) within math/code experiments respectively, to ensure a fair comparison.
>
> Other hyper-parameters such as $k$ are ablated in the experiments. We bump the $k$ for code experiments since CodeContests is a challenging benchmark.

---

> > ### Comment · Reviewer_xLpG · 2025-04-02
> >
> > Thanks for the responses! I have some follow-up questions regarding the responses:
> >
> > **General RL comparison**
> >
> > It is good to see the generalization across pass@k with different k values, but my concern about separate training objectives is **whether training the model with GRPO + pass@k can outperform GRPO + pass@1, (correct me if I am wrong, where the former is the method proposed in the paper, and the latter is the classical approach), when the evaluation metric is maj@p**, and vice versa. Any other algorithms are also acceptable since only the objective matters here.
> >
> > **Evaluation benchmark**
> >
> > I encourage the authors to evaluate the method on MATH500, which is derived from the original MATH dataset [1] to avoid potential contamination issues. During the development of PRM800K [2], since the initial 7.5K training set was insufficient for training a robust Process Reward Model (PRM) on step-by-step solution data, 4.5K problems from the test set were incorporated into the training set, leaving a remaining subset of 500 problems now referred to as MATH500. Since the release of PRM800K, MATH500, instead of the original MATH test set, has been widely adopted to prevent overlap between training and test sets. Also, datasets like AIME2024 and AIME2025, which are even less likely to be contaminated, would be better options for evaluation.
> >
> > **Clarification**
> >
> > Can the author point me to the ablation/evidence show that “For training, the experiments are rather robust to the choice of learning rate and other training parameters”?
> >
> > **Hyperparameter and base model selection (optional)**
> >
> > Moreover, the authors might want to tune the hyperparameters following the recipes in [1]; based on some reproduction work, temperature=1 doesn't give the desired model performance. Additionally, studies like [2] reveal that running outcome reward RL on top of Llama produces more noise and less appealing performance. The authors could also consider including results on other base models if time permits.
> >
> > **Reference**
> >
> > [1] Hugging Face, . "Open R1: A fully open reproduction of DeepSeek-R1." (2025).
> >
> > [2] Gandhi, Kanishk, et al. "Cognitive behaviors that enable self-improving reasoners, or, four habits of highly effective stars." arXiv preprint arXiv:2503.01307 (2025).
> >
> > [3] Hendrycks, Dan, et al. "Measuring mathematical problem solving with the math dataset." arXiv preprint arXiv:2103.03874 (2021).
> >
> > [4] Lightman, Hunter, et al. "Let's verify step by step." The Twelfth International Conference on Learning Representations. 2023.
> >
> > [5] Guo, Daya, et al. "Deepseek-r1: Incentivizing reasoning capability in llms via reinforcement learning." arXiv preprint arXiv:2501.12948 (2025).
> >
> > [6] OpenAI. “Learning to reason with LLMs.” url: https://openai.com/index/learning-to-reason-with-llms/ (2024)

---

> > > ### Author Response · Authors · 2025-04-02
> > >
> > > We thank the reviewer for engaging with our rebuttal. We agree that addressing the concerns raised (eval on MATH500 and AIME24, 25 for example) will strengthen the paper. The issues pointed out are all omissions (of results, baselines, etc.) due to the rapid evolving landscape in the RL on LLM field (given that the manuscript is submitted in January and we are, same as the reviewer, quite aware of the evolved references here happening in Feb and March; techniques like GRPO, the PPO variant by using MC rollouts to estimate the value $V$, gained prominence only recently with the release of Deepseek-R1 in mid-Jan), rather than flaws in the method derived therectically from the optimization objective or experimental design for empirical evidence, and since most of them can be addressed easily we would like to kindly ask the reviewer to consider raising their score if they feel the solutions discussed below are satisfactory.
> > >
> > > > General RL comparison
> > >
> > > If we train with pass@k and evaluate with maj@k, there is generally no guarantee that we can do better than pass@1.
> > >
> > >  However, our work targets situations where the inference time objective is known to inform the training time procedure. We need to make assumptions, this is the trade-off. If we have no assumptions on the inference time objectives, there is in principle no better way than training with pass@1.
> > >
> > > > Evaluation benchmark
> > >
> > > We agree it is a valid point to measure contamination especially when we want to report SoTA numbers that compare against prior results.
> > >
> > > However, in our case we believe it is more meaningful to measure the "delta" of performance, since the improvements are compared across different methods.
> > >
> > > AIME 24 and AIME 25 are fair benchmarks too, but since we do not claim SoTA on reasoning benchmarks and our contribution rests on measuring the relative performance improvements across different methods, having MATH and HARP dataset should provide valid empirical evidence as well.
> > >
> > > > Clarification
> > >
> > > We do not have results in the paper showcasing this, though in very early experiments we settle with a learning on which the baseline methods (pass@1) work stably. We hence use the same lr for the pass@k objectives too, and have not changed the lr since.
> > >
> > > Does the reviewer believe that we should provide ablation on lr too? Learning rate, among many other hyper-parameters in the system, can in principle all be ablated and realistically it is infeasible to ablate across all combinations. We have ablations for important hyper-parameters to the problem such as the value of k.
> > >
> > > > Hyperparameter and base model selection (optional)
> > >
> > > We agree that temp=1 is not necessarily the best sampling config for max'ing out evaluation performance - in fact in our experience greedy sampling might add a few points to the evals. That said, we feel that using temp=1 and top-p=1 ensure that training and eval use the same sampling config (temp=1 and top=1 is on-policy sampling for RL). This is meant to reduce confounding factors introduced by the discrepancy between eval and training.
> > >
> > > Other base models: we do not mean to pursue SoTA results in this work (as data and the starting model plays a crucial role here), and rather we feel it is more meaningful to measure the "delta" in performance across methods to provide a sound scientific ground for algorithmic advancement. Though the absolute numbers might not transfer across different models (llama vs. deepseek r1), we believe the "delta" should hopefully have a better transferable property.

---

### Official Review · Reviewer_61s6 · 2025-03-16

**Overall Recommendation:** 3

**Summary:**

This paper investigates the reinforcement learning algorithms of LLMs in the training time for achieving the test-time objectives. Specifically, it focuses on k-sample policy gradient approaches assuming pass@k and majority vote are the test-time strategies of interest. The authors proposed a leave-one-out like advantage function, which is claimed to be an unbiased estimation of the policy gradient. Furthermore, they showed another variant which further reduces the variance through introducing a bias term by adding an average baseline in the advantage function. Empirical results showing that the biased k-sample method outperforms standard policy gradients. The proposed algorithm is validated through different benchmark datasets (e.g., MATH, CodeContests and TACO) using different sizes of models, demonstrating practical applicability of the proposed methods.

**Claims And Evidence:**

Yes.

**Essential References Not Discussed:**

No.

**Experimental Designs Or Analyses:**

Yes.

**Methods And Evaluation Criteria:**

Yes.

**Other Comments Or Suggestions:**

- I suggest the authors add a complete proof for lemmas in the appendix for better understanding.

**Other Strengths And Weaknesses:**

## Other Weaknesses
- The variance reduction is not shown both theoretically and empirically.
- The only baseline the authors considered was mean policy gradient, which is not comprehensive.
- The comparisons/inconsistent results on PPO integration are not explained adequately. For example, why does pass@k policy gradient become much worse than the baseline when PPO is used?

**Questions For Authors:**

- What is the performance comparison between the proposed approach (e.g., pass@k policy gradient) and purely SFT trained or pass@1 mean policy gradient method with pass@k test-time strategy? This would be meaningful as it shows how much benefit the proposed method can bring.

**Relation To Broader Scientific Literature:**

The contribution of this paper has shown that aligning the test-time objective with the training-time effort is promising, which is related to RL training and test-time compute of LLMs.

**Theoretical Claims:**

- In the proof of Lemma 1, why is the expectation of two independent variables zero? I didn’t see any assumption made on either variable being zero.
- The proof for Lemma 2 is hard to follow. Not sure which expression the authors refer to. I suggest the authors add a complete proof in the appendix for better understanding of the claims.

---

> ### Author Rebuttal · Authors · 2025-03-31
>
> We thank the reviewer for the time and efforts in providing us with the valuable reviews. We will incorporate your feedback into our later revision. We address your comments below.
>
> > In the proof of Lemma 1, why is the expectation of two independent variables zero? I didn’t see any assumption made on either variable being zero.
>
> This is because the expectation of the score function is zero $E[ \log \pi(y_i|x)]=0$. More precisely,
>
> $E[f_{-i}\log \pi(y_i|x)] = E[f_{-i}] \cdot E[\log \pi(y_i|x)]=0$
>
> Where the equality makes use of the statistical independence between $i$ and other samples $-i$. We will make this more clear in the revision.
>
> > The proof for Lemma 2 is hard to follow. Not sure which expression the authors refer to. I suggest the authors add a complete proof in the appendix for better understanding of the claims.
>
> Thank you for the suggestion, we will add a more detailed explanation in the Appendix.
>
> The goal of Lemma 2 is mainly to derive the analytic form of the biased gradient estimate, which turns out to optimize a biased objective with an interesting analytic form - it is the average of leave-one-out objectives from $k$ samples.
>
> > The variance reduction is not shown both theoretically and empirically.
>
> The understanding that the control variate improves variance, either in the form of the leave-one-out control variate (for the unbiased estimator), or subtracting a mean baseline (for the biased estimator), was based on common consensus in the literature on this topic [1]. In fact, we cannot show theoretically that the variance is reduced in all cases, due to the multiplication with the score functions; but we can show that the advantage estimation has lower variance. We can make the theoretical argument more precise in the revision.
>
> For empirical validation: we never implemented an estimator without control variate for variance reduction, as such estimators are plain REINFORCE gradient estimators and do not work well in practice. We can add such an ablation in the revision in case the reviewer finds it useful.
>
> [1] Kool et al, Buy 4 REINFORCE Samples, Get a Baseline for Free!, 2023
>
>
>
> > why does pass@k policy gradient become much worse than the baseline when PPO is used?
>
> We have some speculations regarding this observation: the pass@k policy gradient, compared to the mean policy gradient and biased pass@k policy gradient, has the sparsest gradient, i.e., the weight before the policy gradient is non-zero only for positive samples where it is the only 1 positive sample among k samples, while for the other samples the weight is 0. Biased pass@k policy gradient, by subtracting the mean among k samples, makes the policy gradient no-sparse in such cases. When incorporating PPO, adding a learnt value breaks this sparsity due to the imperfectness of the value model.
>
> > What is the performance comparison between the proposed approach (e.g., pass@k policy gradient) and purely SFT trained or pass@1 mean policy gradient method with pass@k test-time strategy? This would be meaningful as it shows how much benefit the proposed method can bring.
>
> We’ve observed a performance drop when doing SFT on CodeContests training set. Given that Llama 3.1 70B has already been heavily tuned in the post-training, some code
> solutions in CodeContests training set are of less quality than the data presented in its
> post-training phase. For example, some imports in the Python codes are outdated (e.g., from
> fractions import gcd will throw an ImportError since Python 3.9).
>
> Therefore, we included a baseline for Rejection Sampling Finetuning (RFT) instead, where the model generates 200 rollouts for each problem instance in the training dataset and collect the correct solutions, we downsample them to maximum 50 solutions per problem and conduct SFT on this dataset.
>
> We show the comparison below, the best performance is bold and the second best is underscored. With these comparisons, the both pass@k (k=8) variants achieves the best performance in terms of the closest metrics pass@10.
>
> | Method                         | pass@1 | pass@10 | pass@100 |
> |---------------------------------|--------|---------|----------|
> | **Llama 3.1 70B Inst.**         | 16.1   | 34.2    | 48.2     |
> | **SFT**                         | 10.0   | 26.7    | 39.2     |
> | **Rejection Sampling Finetuning**| 18.1   | 37.2    | 52.5     |
> | **mean PPO**                    | 24.9   | 34.6    | 41.1     |
> | **pass@k PPO**                  | 21.2   | 41.4    | 51.0     |
> | **biased pass@k PPO**           | 22.6   | 45.1    | 56.3     |

---

> > ### Comment · Reviewer_61s6 · 2025-04-04
> >
> > Thanks for the response, but my concerns have not been fully addressed. Additionally, I want to know:
> > - How would the expectation of the score function be zero? It is just log probability of tokens, which seems to me should be positive.
> > - Why does the new result suggest that the pass@k training does not generalize well to other metrics? For example, the gain of biased pass@k PPO is much smaller on pass@100 compared with Rejection Sampling Finetuning (RSF). Not to mention that pass@k PPO is even worse than RSF.
> >
> > I found the setting of the paper interesting, but it may need some revisions to improve the presentation and results. In light of this response, I will keep my score.

---

> > > ### Author Response · Authors · 2025-04-05
> > >
> > > Thank you for engaging with our rebuttal, we appreciate your time very much.
> > >
> > > > How would the expectation of the score function be zero? It is just log probability of tokens, which seems to me should be positive.
> > >
> > > We apologize for this confusion - the score in our case is at times -1 for failure and +1 for success. Hence, a negative score means that the model has not been able to answer correctly >50% of all questions in the eval set or training set. The non-negative score - which is the accuracy of the evaluation - can be obtained by a simple linear transformation. We will make this more clear in the revision.
> > >
> > > > Why does the new result suggest that the pass@k training does not generalize well to other metrics? For example, the gain of biased pass@k PPO is much smaller on pass@100 compared with Rejection Sampling Finetuning (RSF). Not to mention that pass@k PPO is even worse than RSF.
> > >
> > > We speculate there are some good technical reasons for this, which are insightful for practitioners and researchers in general.
> > >
> > > In our Rejection Sampling Finetuning (RSF) setting, the number of generations is 200 per problems for the whole training set (13k problems) so in total 2.6M trajectories are sampled. In contrast, in our PPO experiments (results from Table 1) setup detailed in A.4, a total  number of 512000 trajectories (8000 gradient steps * 2 batches / steps / sampler * 32 sampler) are sampled.
> > >
> > > We think in some cases, RSF can have an edge over PPO because it leverages offline samples at larger scale and makes use of off-policy data and offline data in a deliberate way, which might have a positive impact. PPO, as well as many other online RL algorithms, is known for shrinking the policy entropy over training due to online training, and as a result limits the model's exploration ability over time. This somehow bounds the performance, at the cost of being more data efficient (512k samples v.s. 2.6M samples). The RFT, a offline setting, enjoys a lot the exploration (able to find the correct solutions for hard problems if it's solved once out of 200 attempts) and benefits from the subsequent filtering we set, while it takes more compute to obtain the total samples.
> > >
> > > We will make this discussion more clear in the revision. In the meantime please let us know if you have other technical questions you'd like more clarification on.

---

### Official Review · Reviewer_dbZK · 2025-03-21

**Overall Recommendation:** 2

**Summary:**

This work presents a training objective that directly optimizes pass@k and maj@k performance for LLMs, and optimizes them using standard RL training algorithms (policy gradients with a baseline, PPO) known in practice. They evaluate performance on a synthetic bandit task, math reasoning and coding tasks, and find that they are able to improve performance over algorithms that only optimize pass@1 performance.

## update after rebuttal
In the LLM literature, it is well-established that optimizing for pass@1 does not necessarily improve pass@k. Thus, it makes sense to optimize it with an RL training algorithm like Reinforce. The main contribution of the paper is to apply Reinforce on the pass@k objective. The results and discussion are interesting. My main concern is with technical novelty since prior works like Chow et. al. do exactly the same thing (choice of optimizing pass@n for a trained vs. ground-truth reward function does not change the algorithm or setup). **But, I am still willing to raise my score to 2.5 and would not be strongly opposed to accepting the paper if other reviewers are willing to champion it, since the empirical insights are useful to the community**.

**Claims And Evidence:**

Yes, the claims made are clear. They propose an objective to optimize pass@k directly and run experiments that implement this objective and RL algorithm.

**Essential References Not Discussed:**

No, I believe the paper discusses most relevant works that optimize pass@n performance. Though, in the next iteration, it would be useful to include some discussion on training time interventions, that optimize for inference time performance, for example more recent meta-RL training objectives like Optimizing Test-Time Compute via Meta Reinforcement Fine-Tuning, Qu. et. al 2025.

**Experimental Designs Or Analyses:**

Yes, I read through their experimental setup for math reasoning and coding benchmarks, and it looks reasonable to me.

**Methods And Evaluation Criteria:**

Yes, the math and coding benchmarks used seem standard. The authors only plot performance on the pass@k objective they trained for. It would be interesting to see, if the performance gain on value of k used in training also extends to other values of k. If not, how was this value of k (4 or 8) chosen in practice? This part was a bit unclear.

**Other Comments Or Suggestions:**

- If the authors can run experiments against suggested baselines that would strengthen the paper.
- If the authors can also demonstrate that the approach can extend to broader functions like maj@k or even self-verification@k (see Sample, Scrutinize and Scale: Effective Inference-Time Search by Scaling Verification, Zhao et. al.), that would also strengthen this paper.
- In general if the authors can empirically show that the proposed approach can improve performance on math reasoning, either at a broader range of KL divergence, or for a broader range of k (beyond the pass@k trained for), that would be great.

**Other Strengths And Weaknesses:**

Strengths:
- The gains on pass@8 performance for coding benchmarks is impressive, compared against a baseline that only optimizes pass@1.
- The proposed approach is fairly straightforward to implement.

Weaknesses
- The paper lacks key comparisons with InfAlign (Balashankar et. al.), and Chow et. al., that both propose RL training objectives that directly optimize pass@n or BoN performance.
- The performance on MATH is underwhelming, especially Maj@k in Figure 15, and even pass@4 in Figure 2, is only better in a narrow KL regime, with the proposed policy gradient.

**Questions For Authors:**

- Do the authors think that there is a fundamental tradeoff between optimizing for pass@1 and pass@k performance? Why is it that we cannot learn a single model with good pass@1 performance on easy problems and good pass@k on hard problems? Is this lack of representation capabilities, pre-training biases, or issues with training objectives/algorithms?

**Relation To Broader Scientific Literature:**

The key idea in this paper is to optimize for inference time performance, like pass@k or maj@k. From prior work, we know that optimizing for pass@1 may not necessarily lead to better performance on pass@k. So, the idea is to improve performance at a higher compute budget. In general, the broader goal should be to learn policies that push up the pareto frontier of performance against inference time compute. This work does not do that, but instead makes an attempt to push the performance up at higher test compute budget, i.e., optimize for performance at higher pass@n.

**Theoretical Claims:**

The only theoretical claims are made with respect to the bias of their proposed gradient estimates, which seem correct given their readily apparent nature.

---

> ### Author Rebuttal · Authors · 2025-03-31
>
> We thank the reviewer for the time and efforts in providing us with the valuable reviews. We will incorporate your feedback into our later revision. We address your comments below.
>
> > The paper lacks key comparisons with InfAlign (Balashankar et. al.), and Chow et. al., that both propose RL training objectives that directly optimize pass@n or BoN performance.
>
> We have discussed the technical differences between our work and Balashankar et. al, Chow et al, in Section 3.3, with more extended discussion in Appendix C.
>
> As pointed out in the paper, Balashankar et. al and Chow et al consider structurally different problem formulation as we have in this work. Balashankar et. al considers a pairwise preference setting (vs. point-wise reward setting in our case) while Chow et al considers the case with auxiliary models such as verifiers or scorers (vs. no requirement for auxiliary models in our case). As a result, it is challenging to set up an apple-to-apple comparison against such concurrent work since they tackle very different settings. Further, we also feel that it’s fair not to make direct comparison to unpublished concurrent work.
>
> > The performance on MATH is underwhelming, especially Maj@k in Figure 15, and even pass@4 in Figure 2, is only better in a narrow KL regime, with the proposed policy gradient.
>
> We acknowledge the fact that the performance improvement on MATH is not huge, though they are still statistically significant. The upper bound of the performance improvement also depends heavily on the base model we started with (which we do not have full control over). MATH as a benchmark task is also relatively easy since the pass@16 has also reached ~86%, at a competitive level as the frontier models such as O1 and Deepseek R1. This also puts a limit on the amount of improvement.
>
> The narrow KL divergence regime: Fig 2 shows improvement across all KL values we observe during training. For Fig 15, since maj@k signals are much sparser, this makes it more difficult for the model to reach a larger KL divergence deviation during training. However, for the KL divergence regime it reaches, the performance improvements are clear. We have highlighted this point as an algorithmic trade-off in the paper.
>
> Additionally, depending on practical applications, the “narrow KL” regime might suffice since for general post-training, ideally the RLHF stage does not lead to too much deviation from the reference policy. We leave this to the judgement of practitioners.
>
> > Do the authors think that there is a fundamental tradeoff between optimizing for pass@1 and pass@k performance?
>
> In principle, pass@1 and pass@k share common goals but there is a conflict between the two objectives, in that pass@k always encourages higher diversity and coverage over the solution space, while pass@1 encourages more determinism.
>
> > Why is it that we cannot learn a single model with good pass@1 performance on easy problems and good pass@k on hard problems? Is this lack of representation capabilities, pre-training biases, or issues with training objectives/algorithms?
>
> It should be possible in principle: this might require a hybrid objective, and switching between pass@1 vs. pass@k training objectives depending on the problem difficulty. We might even want to devise a novel algorithm that adjusts the objective based on the online sampled pass rate. This definitely is a promising path for future work that integrates the two objectives into a unified algorithm.
>
> The pretraining biases and representation capabilities certainly plays a key role as well, as suggested by recent investigations to reproduce Deepseek R1 (e.g., Qwen models can reproduce R1 behavior more consistently, and model sizes matter too). That said, training algorithms / objectives should make a difference, as we have demonstrated in the paper, especially because RL has become increasingly important in post-training.

---

> > ### Comment · Reviewer_dbZK · 2025-04-05
> >
> > Thank you for the detailed response, especially for the clarifications on the poor performance for Maj@k, and gains in a narrow KL regime. Those concerns are addressed. But, I still think that the "Chow et al.",  setting is very similar to this paper, in that even they optimize for "pass@k on math". I believe both your work and theirs assumes access to ground-truth outcome rewards, so it is definitely a valid baseline. While I agree that this work is not published, it was certainly available on arXiv in 2024, and warrants a comparison, *even a single dataset/experiment comparing the two approaches would be helpful*. I will keep my score for now, and update (if needed) after discussions with other reviewers and AC.

---

> > > ### Author Response · Authors · 2025-04-05
> > >
> > > Thank you for engaging with our rebuttal, we appreciate your time very much.
> > >
> > > > But, I still think that the "Chow et al.", setting is very similar to this paper, in that even they optimize for "pass@k on math". I believe both your work and theirs assumes access to ground-truth outcome rewards, so it is definitely a valid baseline.
> > >
> > > The key difference is that: Chow et al assumes access to a verifier score $r$, which it has access to at both training and inference time. In our case, we assume access to the ground truth score, but only at training time.
> > >
> > > That said, both our method evaluates with pass@k at evaluation time - though Chow et al trains with best-of-k with the verifier $r$ strategy at training time, while we train with pass@k.
> > >
> > > > While I agree that this work is not published, it was certainly available on arXiv in 2024, and warrants a comparison, even a single dataset/experiment comparing the two approaches would be helpful.
> > >
> > > We are happy to provide a more detailed discussion of the difference + a single experiment comparison to Chow et al, in the revision. We can simulate Chow et al's setting for verifier with a corrupted ground truth score, and see its impact on the evaluation performance.
> > >
> > > Please let us know if you have other questions that you'd need clarification on to adjust our scores, thanks.

---

### Official Review · Reviewer_KzmQ · 2025-03-23

**Overall Recommendation:** 2

**Summary:**

This paper explores the potential of explicitly optimizing language models for inference-time performance objectives, particularly pass@k and majority voting, using reinforcement learning (RL). The authors propose a k-sample objective formulation and derive both unbiased and biased gradient estimators, including a leave-one-out control variate for variance reduction. The paper provides theoretical justification for the estimators, including proofs of unbiasedness and characterizations of induced biases. Empirical evaluations on mathematical reasoning (MATH, HARP) and code generation (CodeContests, TACO) benchmarks demonstrate that training-time optimization of inference-time metrics can yield superior performance on those metrics (especially pass@k), compared to standard mean policy gradient approaches.

**update after rebuttal**:
Thanks for the authors' responses. My concern remains that the generalization sometimes degrades when optimizing inference-time metrics. However, the overall idea and method are interesting, and I encourage the authors to explore more on this.

**Claims And Evidence:**

Claims:
Inference-time objectives like pass@k and majority voting can be explicitly optimized during training. Doing so leads to improved performance on those inference-time metrics, especially on more difficult problems. The proposed gradient estimators are theoretically justified and practically effective.

While the training-time improvements are consistent, test-time generalization improvements are more mixed, especially in the majority voting case and for high-capacity models (e.g., LLaMA 70B on MATH). The authors acknowledge this.

**Essential References Not Discussed:**

The related work section (and appendix) is comprehensive and cites relevant prior work on:
- RL with multi-sample objectives,
- Policy gradient variance reduction,
- Self-consistency and majority voting.

**Experimental Designs Or Analyses:**

The experimental design is sound and robust. Key strengths:
- Ablations on different model sizes (3B, 8B, 70B).
- Evaluation across multiple benchmarks and multiple difficulty levels.
- Consideration of both training performance and test-time generalization.
- Integration of the proposed gradient estimators with PPO, demonstrating compatibility with standard RL fine-tuning pipelines.

One weakness is the limited analysis of why generalization sometimes degrades when optimizing inference-time metrics. This could benefit from further exploration.

**Methods And Evaluation Criteria:**

The methods are clearly described and well-motivated. The paper adapts standard reinforcement learning techniques to k-sample objectives, and proposes novel unbiased and biased gradient estimators for these settings. Evaluation is performed on realistic and challenging benchmarks: MATH and HARP for mathematical reasoning, and CodeContests and TACO for code generation.

Metrics are appropriate for each domain. Evaluation is comprehensive, including training curves, sample-efficiency analysis (via KL-divergence plots), and ablations on model size and number of samples k.

**Other Comments Or Suggestions:**

- Consider emphasizing earlier that these methods do not require auxiliary models (unlike best-of-k with reward models)
- Including a discussion on compute efficiency (vs benefit) can help practitioners decide when to use these techniques.

**Other Strengths And Weaknesses:**

Strengths:
- Good theoretical and empirical contributions.
- Clearly written, with well-illustrated examples and intuitions.

Weaknesses:
- Generalization of improvements to evaluation time is not always consistent.
- The method may be less effective for problems where models already achieve high performance (e.g., MATH with LLaMA 70B).

**Questions For Authors:**

- Can you elaborate on why training-time improvements on pass@k/majority voting objectives do not always translate to better test-time performance? Could this relate to diversity collapse or overfitting to training prompts?
- When should practitioners prefer biased vs unbiased gradient estimators? Do you recommend any heuristics for setting k?

**Relation To Broader Scientific Literature:**

The paper is well-positioned in the context of:
- RLHF and preference-based fine-tuning.
- Best-of-k and self-consistency generation
- Inference-time optimization

**Theoretical Claims:**

The paper contains several theoretical claims, mainly regarding gradient estimation:
1. The leave-one-out gradient estimator is unbiased.
2. The biased variant optimizes a modified leave-one-out objective.

These derivations follow standard techniques in policy gradient literature. The proofs are concise, correct, and help contextualize the practical impact of bias vs variance.

---

> ### Author Rebuttal · Authors · 2025-03-31
>
> We thank the reviewer for the time and efforts in providing us with the valuable reviews. We will incorporate your feedback into our later revision. We address your comments below.
>
> > 1. Generalization of improvements to evaluation time is not always consistent.
>
> The generalization of improvements to evaluation is indeed less straightforward, as we have explicitly discussed in the paper. The generalization gap depends in a complex way on the training data and model sizes. The cleanest way to assess our algorithmic progress is via the training performance, though we have also included evaluation for completeness. We still see evaluation improvements overall, though they are not as strong as training rewards improvements.
>
> >  2. The method may be less effective for problems where models already achieve high performance (e.g., MATH with LLaMA 70B).
> This is true and we have elaborated on this observation in the paper. The k-sample objectives are generally less effective for easy problems to the model, since the model does not have strong signals to make improvements. However, for more difficult problems (HARP for Llama 70B), we see more significant improvements, which might be a more useful case for many practitioners as well.
> > Consider emphasizing earlier that these methods do not require auxiliary models (unlike best-of-k with reward models)
>
> We will make sure to emphasize this point earlier on in this work. We have introduced the inference objectives of interest in Section 2.
>
> > Including a discussion on compute efficiency (vs benefit) can help practitioners decide when to use these techniques.
>
> We will elaborate on such points more extensively in the revision. In general, we find that the newly proposed algorithms automatically improve inference time objectives with a similar computational budget as the baseline algorithms (which maximizes the mean objectives). The algorithmic change is also almost a few-line code change, which is quite convenient for practitioners.
>
> > Can you elaborate on why training-time improvements on pass@k/majority voting objectives do not always translate to better test-time performance? Could this relate to diversity collapse or overfitting to training prompts?
>
> We conjecture that this might be more related to the generalization gap between training and test set, which depends in a rather complex way on factors such as the model size, and training data that went into the model pre-training and SFT (the released Llama 3.1 model might have SFT’ed on similar dataset to our RL prompt set, which might reduce the level of generalization we expect). As a result, maybe training set diversity and performance improvements do not transfer 100% to the evaluation set.
>
> > When should practitioners prefer biased vs unbiased gradient estimators? Do you recommend any heuristics for setting k?
>
> We feel that a good approach is to decide on the estimator adapting to the specific use case. Overall, we find the two estimators to be both quite competitive, and since the unbiased gradient estimator optimizes for the exact objective, it might be the first thing to try.
>
> For setting k: this depends on the application of interest. We can choose a specific k that will be used at inference time. We can also choose a generic value of k to maximize model performance while diversifying the samples as much as possible - in general, a large value of k indicates more diversity. We find that values of {4,8,16} are good starting points, depending on the application.

---

### Decision · Program_Chairs · 2025-05-01

**Decision:**

Accept (poster)

**Comment:**

Language models are often optimized for next-token prediction but later evaluated on downstream tasks using metrics such as pass@k. This paper introduces a novel reinforcement learning technique that directly optimizes pass@k. According to reviewers, the paper is well-written and offers both theoretical and experimental contributions. Although there are concerns regarding two concurrent works, the authors address them in the related work section and explain the key differences. I recommend to discuss these reference in the introduction and provide experimental comparisons with at least one of them.